# Probabilistic and Machine Learning Methods for Uncertainty Quantification in Power Outage Prediction due to Extreme Events

Prateek Arora[1] and Luis Ceferino[1,2]

[1]Civil and Urban Engineering Department, New York University, Brooklyn, NY, 11201, USA
[2]Center for Urban Science and Progress, New York University, Brooklyn, NY, 11201, USA

**Correspondence:** Prateek Arora (prateek40.a@gmail.com)

**Abstract.** Strong hurricane winds damage power grids and cause cascading power failures. Statistical and machine learning models have been proposed to predict the extent of power disruptions due to hurricanes. Existing outage models use inputs including power system information, environmental, and demographic parameters. This paper reviews the existing power outage models, highlighting their strengths and limitations. Existing models were developed and validated with data from a few utility companies and regions, limiting the extent of their applicability across geographies and hurricane events. Instead, we train and validate these existing outage models using power outages from multiple regions and hurricanes, including Hurricanes Harvey (2017), Michael (2018), and Isaias (2020), in 1,910 U.S. cities. The dataset includes outages from 39 utility companies in Texas, 5 in Florida, 5 in New Jersey, and 11 in New York. We discuss the limited ability of state-of-the-art machine learning models to (1) make bounded outage predictions, (2) extrapolate predictions to high winds, and (3) account for physics-informed outage uncertainties at low and high winds. For example, we observe that existing models can predict outages higher than the number of customers (in 19.8% of cities with an average overprediction ratio of 5.2) and cannot capture well the outage variance for high winds, especially above $70\,m/s$. Finally, we suggest that a Beta regression outage model could address these current shortcomings.

## 1 Introduction

Hurricanes can cause significant damage to the power distribution systems resulting in large power failures and losses of billions of U.S. dollars (Smith, 2020). Strong winds from hurricanes can destroy the exposed overhead distribution lines in a power grid and cause cascading power failures. For example, Hurricane Isaias (2020) damaged old power infrastructure and caused more than two million power outages across the U.S. More than a million outages occurred in New Jersey (https://www.nytimes.com/2020/08/04/nyregion/isaias-ny.html, last access: 21 September 2022) even though Hurricane Isaias had transitioned to a tropical storm when it hit New Jersey, reducing its sustained winds to 25 m/s (Latto et al., 2021). To address this issue, the U.S. Department of Energy (DOE) has prioritized investing in enhancing power infrastructure resilience (National Academies of Sciences, Engineering, and Medicine, 2017). The Senate of the U.S. passed the Grid Research Security Research and Development Act (2020) with a budget of 573 million U.S. dollars to be spent from 2020-2025 to improve grid security to withstand shocks and rapidly recover from disruptions (Congress.gov, 2020).

Hurricane induced power interruptions can cause billions of dollars in losses and long-lasting impacts on vulnerable communities. The power outages caused by storms can last for several hours to weeks and even months (https://www.reuters.com/business/environment/over-397000-still-without-power-florida-after-hurricane-ian-2022-10-04/, last access: 16 February, 2023). Large-scale power blackouts show the vulnerability of the power grid to hurricanes, e.g., (1) 8.1 million homes lost power during Superstorm Sandy (2012) (Sheppard and DiSavino, 2012), (2) 1.7 million consumers in the Southeast United States lost power in the aftermath of Hurricane Michael (2018) (EIA.GOV, 2018) and (3) Hurricane Ida (2021) was responsible for 1.2 million electrical outages (AJOT, 2021). Critical infrastructure systems such as hospitals and fire departments are especially vulnerable since they need to have the power restored within a few hours after a power outage to respond to the disaster (Ceferino et al., 2018, 2020).

Utilities must first assess the vulnerabilities in their power system infrastructure to enhance their resilience to hurricanes. Researchers have developed machine learning models to help utilities evaluate their vulnerabilities to predict the extent of power outages from hurricanes. These outage models use inputs including hurricane winds, power systems, environmental, and demographic information. Outage prediction models can assist utilities in planning and placing their resources before and during an extreme event for an emergency response to rapidly recover the failed power distribution systems (Arab et al., 2016). These models can also inform about the existing vulnerabilities so utilities can also plan for grid hardening before a hurricane damages the power grid (Ouyang and Dueñas-Osorio, 2014).

Liu et al. (2005) developed a Negative Binomial Generalized Linear Model (GLM) to predict the power outages in North and South Carolina. This model used hurricane parameters such as maximum wind speeds and duration of wind speeds over $20\,m/s$; environmental parameters including land cover, tree type, soil drainage properties, precipitation; and utility information on the number of transformers and customers. The model included only a specific utility, which limited the use of the outage model to other regions. The model also included a storm indicator, making the model not applicable to hurricanes with different characteristics than the ones in the training data. Liu et al. (2007) also presented an Accelerated Failure Time model to estimate the power outage duration. Next, Liu et al. (2008) investigated the spatial correlation of power outages through spatial Generalized Linear Mixed Models (GLMM) but did not observe any significant improvements in power outage prediction.

Han et al. (2009b) also developed a Negative Binomial GLM to predict outages in the Gulf Coast based on extensive information on hurricane parameters, additional environmental indicators (*e.g.,* precipitation, soil moisture, tree type, land cover), and system information (*e.g.,* number of poles, number of transformers). This model did not include any specific utility and storm indicators and instead used only generalizable features (*e.g.,* wind speed, precipitation) to make the model applicable to any hurricanes. Han et al. (2009a) also developed Generalized Additive Models (GAMs) with the same input features as GLMs. GAMs showed an improved accuracy over GLMs in power outage predictions because GAMs can effectively model the highly non-linear behavior of outages and the input parameters, e.g., precipitation and soil moisture can have non-linear effects on power outages (Han et al., 2009a).

Guikema et al. (2010) and Quiring et al. (2011) used decision tree models, Classification and Regression Trees (CART) and Bayesian Additive Decision Trees (BART), with additional topological and soil parameters, to better capture the variability of power outages. Decision trees provide a flexible way to represent the non-linear relation between input parameters and outages.

More recently, researchers developed decision trees-based machine learning methods which are robust to outliers and noise, called Random Forest (Breiman, 2001), to predict power outages caused by storms. Random Forest regression is an extension of decision tree methods for regression. A series of parallel decision trees are fit in the Random Forest regression method to capture non-linearity and achieve high predictive accuracy. Nateghi et al. (2014) calibrated a Random Forest model to outage data from the Gulf Coast. Nateghi et al. (2014) used six input parameters to capture the damaging effects of trees on power

lines. These six input parameters included 3-s gust wind speed, duration of strong winds, soil moisture at different depths, the number of customers served, and tree-trimming practices used to predict outages. Later, Guikema et al. (2014) used only publicly available data to develop a hurricane outage prediction model, independent of utility-specific input parameters, with Random Forest regression using 3-s gust wind speed, strong winds, and the number of customers served.

  Maderia (2015) improved accuracy in power outage predictions with Random Forest models by including information on

tree species. Tonn et al. (2016) used Quantile Regression Forests (Li and Peng, 2011) to predict power outages at different confidence intervals. Guikema et al. (2014) and McRoberts et al. (2018) developed a two-stage zero-inflated power outage prediction model to better account for zero outages. The first stage of such a model is classification to predict outages or no outages. The second stage is the Random Forest regression to predict the count of outages on the point classified as having an outage. Wanik et al. (2017) used a Random Forest model with Lidar-derived tree height data to predict power outages.

Shashaani et al. (2018) developed a three-stage power outage prediction model to improve the accuracy of power outage predictions further. The first stage of the model is a binary classification to predict the location of outages; the second intermediate stage is the clustering of outage locations into a low, moderate, and large number of outages to address high right-skewness of non-zero outage data points; and the third stage is the prediction of the number of outages.

  Previously, researchers have used more complex power outage prediction models, namely neural networks, kernel methods

such as support vector machines, and other tree-ensemble methods, such as AdaBoost, which can model non-linear relationships between input parameters and outages (Xie et al., 2020). Kankanala et al. (2014) employed AdaBoost to predict weather-related power outages. Kankanala et al. (2014) trained a separate model for each city for daily use, and they did not cover extreme weather outages. Eskandarpour et al. (2018); Eskandarpour and Khodaei (2018) used power grid component-level data with support vector machines. Rudin et al. (2012) ranked the power grid components (feeder failures, cables, joints,

terminators, and transformers) based on their vulnerability to extreme weather events. Haseltine and Eman (2017) used a neural network to predict the failure of the power grid components for pre-storm. Such models will require specialized high-resolution power grid component-level data for each city which is not accessible given the data protocols of utility companies. Sun et al. (2018) used Twitter (https://twitter.com; last accessed: 13 January 2023) data to predict real-time outages. Jaech et al. (2018) used repair logs data employing Natural Processing with a Recurrent Neural Network to predict real-time outage durations.

However, tweets (https://twitter.com; last accessed: 13 January 2023) and repair logs are available after the hurricane made an impact on the city. Thus, leveraging repair logs is not possible to predict outages for pre-event planning ahead of a storm. Hence, data availability limits the applicability of these methods at a large scale for power outage predictions from extreme events.

GLM (Liu et al., 2005), GAM (Han et al., 2009a), and Random Forest based power outage prediction models (Guikema et al., 2014; McRoberts et al., 2018; Shashaani et al., 2018) provide outage predictions at a coarser level compared to predictions at component. However, these models are mostly based on open-source, publicly available data and can be generalized at a larger scale to the coastal cities in the United States. Hurricane-caused outages are mostly at the transmission level, which is responsible for city-wide outages (Brown, 2002) (poweroutage.us/faq, last accessed: 13 January 2023) rather than the customer meter level. So, predicting city-wide outages can still guide utilities to arrange for crews and emergency backup power ahead of a storm. Hence, for this paper, we focus on GLM, GAM, and Random Forest based power outage prediction models. As the power outage data is generally not made publicly available by the utilities, the previous models are primarily calibrated to data from a few regions. For example, Liu et al. (2005, 2007, 2008) developed the outage prediction model for North and South Carolina. Guikema et al. (2014), Nateghi et al. (2014), and Shashaani et al. (2018) developed the outage prediction models for the Gulf Coast. This paper addresses this gap by calibrating and validating existing models to extensive outage data from New Jersey, New York, Florida, and Texas at the city level. Thus, we investigate the generalized behavior of power outage models across the United States and focus on publicly available input variables to make our calibrated models widely applicable.

In this paper, Section 2 describes the input features, data sources, and data preprocessing used in the model development for power outage prediction. Section 3 explains the selection of important and uncorrelated input features for model development. GLM, GAM, and Random Forest power outage models are described in sections 4, 5, and 6, respectively. Section 7 describes the results for calibrated models and compares performance with the previous models in the literature. Section 8 highlights the limitations of existing state-of-the-art power outage prediction models to (1) make bounded outage predictions, (2) extrapolate for high winds, and (3) account for physics-informed uncertainties at low and high winds. Section 9 describes the framework for future research to address the shortcomings of existing power outage models, and Section 10 summarizes the findings of this paper.

## 2 Data Description

We acquired power outage data from PowerOutage (poweroutage.us, last access: 21 September 2022), an organization that tracks and records outages from utilities at the city level across the U.S. The automatic outage reporting points of utilities could be hindered during hurricanes which could result in errors in outage counts (poweroutage.us/faq, last access: 13 January 2023). However, PowerOutage (poweroutage.us, last access: 21 September 2022) regularly gets updates from utilities to keep the outage count close to actual outages. The data covered the power outages for Hurricane Isaias (2020) for 11 utilities in New York and 5 in New Jersey, for Hurricane Michael (2018) for 5 utilities in Florida, and Hurricane Harvey (2017) for 39 utilities in Texas. Our dataset has about 3.6 million outages in total. Figure 1 shows outages caused by Hurricane Isaias in New Jersey in 2020. Supplementary Figures S1, S2, and S3 present power outages across New York due to Isaias, Florida due to Michael, and Texas due to Harvey, respectively.

Previously, Liu et al. (2005) developed an outage model at zip code levels and smaller $1km \times 1km$ grid cells. However, the final selected model was zip code level, as the aggregation of input parameters at the $1km \times 1km$ grid level led to more errors

in outage predictions. Since then, outage models have been developed at coarser grids. For example, Han et al. (2009b, a); Guikema et al. (2010); Quiring et al. (2011); Nateghi et al. (2014) developed the models for $2.5km \times 3.7km$ grid cells. Tonn et al. (2016) developed outage models at the zip code level, McRoberts et al. (2018) predicted outages at the resolution of census tracts, and Shashaani et al. (2018) predicted outages at $5km \times 5km$ grid cells.

We calibrated outage models at the city level resolution, comparable to the most recent models by McRoberts et al. (2018) and Shashaani et al. (2018). A model with finer resolution could be developed, provided higher-resolution power outage data and input parameters. Here, we used data with reported outages for 1910 cities in New York, New Jersey, Texas, and Florida. Following the previous literature, we use the covariates listed in Table 1. We obtained model inputs at the city level across all covariates. Further description of the availability, resolution, and methods to obtain each variable at the city level is provided in the subsequent subsections. We also discuss the uncertainties in data that could inherently influence the accuracy of power outage predictions. The total number of data points available is 1910 (cities). The dataset has been divided into train and test datasets with a ratio of 80:20.

## 2.1 Response Variable

We focused on two response variables, the number of outages which is equivalent to the number of customers without power in a city, and the fraction of customers without power. GLM and GAM use Poisson and Negative Binomial distributions to assess the count of outages as they model discrete and non-negative variables. Random Forests can model the fraction of households without power in a city, which is important to compare impact levels across cities.

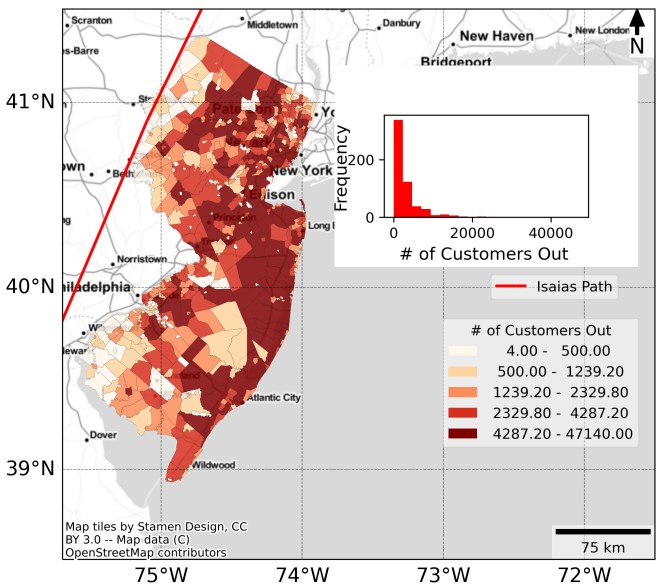

**Figure 1.** Power Outages in aftermaths of Hurricane Isaias (2021) in New Jersey at the city level

## 2.2 Hurricane Parameters

The hurricane parameters considered for this study are 3-s gust wind speed and duration of strong winds over 20m/s, as in previous research (Liu et al., 2005; Han et al., 2009b, a; Guikema et al., 2010, 2014; Shashaani et al., 2018). The overhead distribution system is the most vulnerable component of a power grid to high hurricane winds (National Academies of Sciences, Engineering, and Medicine, 2017). The distribution lines and poles are often close to trees and do not have considerable setbacks. The uprooting of trees due to strong winds often propagates damage to distribution lines. The poles are designed

to sustain wind speeds of around 20m/s (IEEE, 2007). Thus, winds above 20m/s can cause substantial damage to the electric poles. The 3-s gust wind speed and duration of strong winds for the three hurricanes in the dataset were calculated based on a complete wind profile model for tropical cyclones by Chavas et al. (2015). We determined the wind speed for each city at its centroid. Figure 2 illustrates the variation of 3-s wind gusts in New Jersey during Hurricane Isaias. However, wind speed is an important factor causing outages, and any approximation in wind speed estimates could lead to errors in outage predictions. We

provide detailed information in Supplementary Text S1 to demonstrate that wind speeds at the city centroid can be a reasonable estimate in determining the city-wide outages.

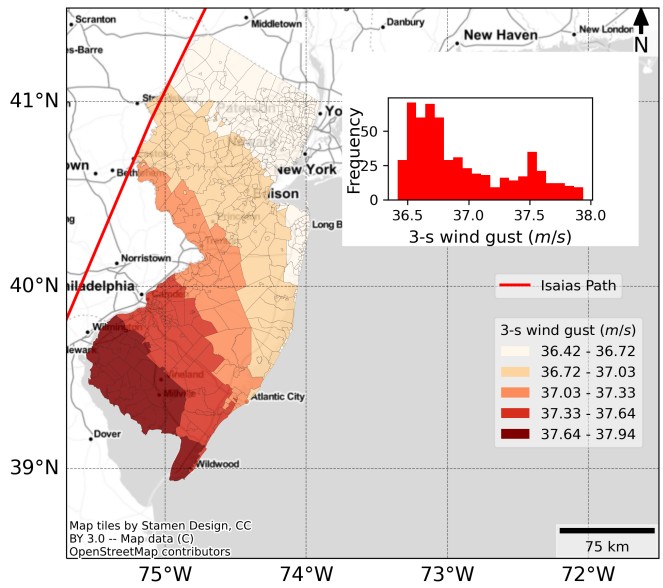

**Figure 2.** Distribution of 3-s wind gust at the city level (Mean: 36.95 m/s, Standard Deviation: 0.41 m/s) across New Jersey before the arrival of Hurricane Isaias

## 2.3 Land Cover Data

Power grid patterns vary for different land use classes, resulting in different outage mechanisms. For example, rural areas can suffer larger power outages since they have radial grid patterns where component failures can propagate more than in

cities with gridded patterns (Petersen, 1982). We obtained National Land Cover Data (NLCD) available from the Multi-

Resolution Land Characteristics Consortium, which is maintained by the United States Geographical Survey (USGS) (https://www.mrlc.gov/viewer/, last access: 21 September 2022). National Land Cover Data has inaccuracies in the thematic pixel classification (Wickham et al., 2021) that could introduce uncertainties in the land cover type. However, evaluating the effect of the inaccuracies in the thematic pixel classification on power outages is not within the scope of this paper. NLCD data is available in raster format with a resolution of 30m x 30m. USGS has classified the original land cover data into 20 different classes. We have reclassified the NLCD data into nine to match previous power outage models. The nine different major classes of land cover data are developed area, water area, barren land, forest area, scrub area, grasslands, pasture land, cultivated cropland, and wetlands. We utilized the spatial analyst in ArcGIS (a tool for Geographic Information Systems) (ESRI 2019) to clip the 30m x 30m land cover raster for each city. We used zonal analysis within ArcGIS to determine the percentage of area covered by the nine major land cover classes.

## 2.4 Precipitation and Soil Moisture Data

Precipitation and soil moisture have been extensively used in power outage models, e.g., Han et al. (2009b); Nateghi et al. (2014); McRoberts et al. (2018). These parameters have non-linear effects on power outages, as deviations below and above standard values can result in more outages. The poles and overhead distribution lines in the vicinity of trees are susceptible to falling trees due to strong hurricane winds. The wet soil conditions from high precipitation and soil moisture increase the likelihood of trees and electric poles uprooting from strong hurricane winds (Han et al., 2009b; Nateghi et al., 2014). Also, persistent drought conditions, e.g., low precipitation in the months before a hurricane, can weaken the roots of trees because of gaps in the soil layer, making trees more susceptible to strong winds (McRoberts et al., 2018).

Precipitation and soil moisture data are available from the variable infiltration capacity (VIC) model from National Land Data Assimilation System Phase 2 (NLDAS2) (Xia et al., 2012; Xia, 2012). However, the limited temporal resolution of parameters required for computing soil moisture and precipitation could introduce errors in the final estimates of these variables (Wei et al., 2013). The limitations of the variable infiltration capacity (VIC) model from the National Land Data Assimilation System Phase 2 (NLDAS2) (Xia et al., 2012; Xia, 2012) to get soil moisture and precipitation are beyond the scope of this study. Precipitation and soil moisture have been recorded each hour since 1979 with a resolution of $0.125° \times 0.125°$. We used nearest-neighbor interpolation to obtain soil moisture and precipitation at the city centroid by taking the value available at the nearest point.

Soil moisture from NLDAS2 is available for three depths, 0-10 cm, 10-40 cm, and 40-100 cm. We calculated daily soil moisture for these depths by taking the average hourly readings. Soil moisture can vary at different geographical locations due to different soil types in different regions. We first normalized soil moisture to compute deviations from average values by computing percentiles. We fit Pearson Type III distributions to the daily time series of soil moisture for all three layers to normalize the soil moisture across different geographies. We use maximum likelihood estimates (MLE) to compute the parameters for Pearson Type III distribution (Hosking and Wallis, 1997). Then, we evaluate the soil moisture percentile. We denote the soil moisture percentiles for three layers of soil at 0-10 cm, 10-40 cm, and 40-100 cm depth as CDF1, CDF2, and CDF3, respectively.

Precipitation data is represented in the form of the Standard Precipitation Index (SPI) (Wu et al., 2007; Guttman, 1998;
Casey, 2016). SPI for the months before the storm's impact can also be used as a proxy for the dryness and wetness of the
soil. For the current study, we calculated SPI for durations of 1 month, 3 months, 6 months, and 12 months by adding hourly
time-series data for precipitation. The following are three steps to compute SPI. First, we fit the Pearson Type III distributions
to the time series of precipitation using MLE. Second, we compute the percentile from the Pearson Type III distribution. Third,
we take the inverse of the calculated percentile using a standard normal distribution to get the SPI for each duration. Figure 3
illustrates the variation of SPI 1 month in New Jersey before the arrival of Hurricane Isaias.

We also included the expected precipitation after the hurricane makes landfall for the next 7 days as heavy rain can lead
to flooding resulting in clustered outages (McRoberts et al., 2018). The soil moisture percentiles and SPI values are obtained
since the day before the hurricane impacts the power systems. This starting time allows for outage predictions to give an early
warning to the utilities and community members and take precautionary steps before strong hurricane winds hit the area.

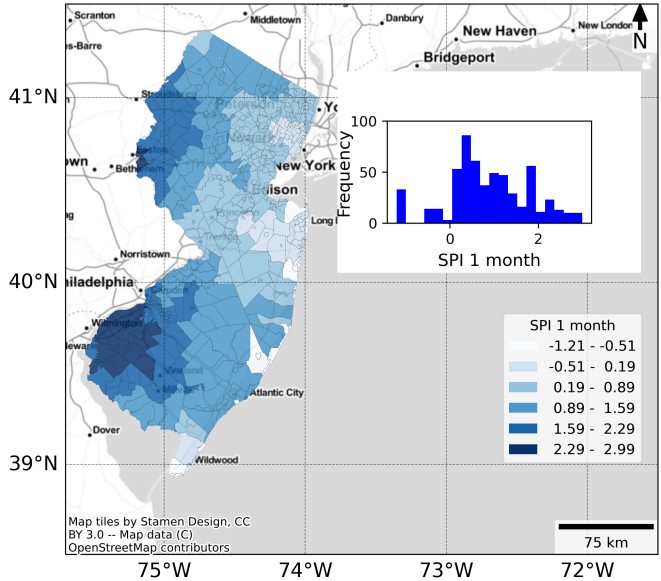

**Figure 3.** Distribution of SPI 1 month (Mean: 0.88, Standard Deviation: 0.93) across cities in New Jersey before the arrival of Hurricane
Isaias

**2.5   Root Zone Depth**

The effective root zone depth is defined as the depth of the soil from which plants and trees can effectively extract water and
nutrients for growth (http://www.wood-database.com, last access: 21 September 2022). The more effective the root zone depth
for trees, the less likely they will fail from strong hurricane winds (McRoberts et al., 2018). We add root zone depth as an
input parameter for outage predictions because it could indicate the hazard from falling trees to the power lines. Root zone data
is available from the United States Department of Agriculture (USDA) under Gridded Soil Survey Geographic (Soil Survey

Staff) at $30m \times 30m$ resolution as raster data. The root zone depth at the city level is calculated as the average of the root zone in a city using the spatial analytic tool in ArcGIS. Given the resolution of available outage data at the city scale, we were not able to consider the variations in root zone depth, which limits the ability of the power outage model to consider the variation of tree root strength within a city.

## 2.6 Percentage Treed Area

USDA created National Insect and Disaster Risk Maps (Krist Jr. et al., 2014) in 2012, with the area covered by trees at $240m \times 240m$ as raster data. The raster tree data is used to calculate the percent of the area covered by trees at the city level using the spatial analytic tool in ArcGIS (ESRI 2019). Figure 4 illustrates the distribution of percent treed area in New Jersey.

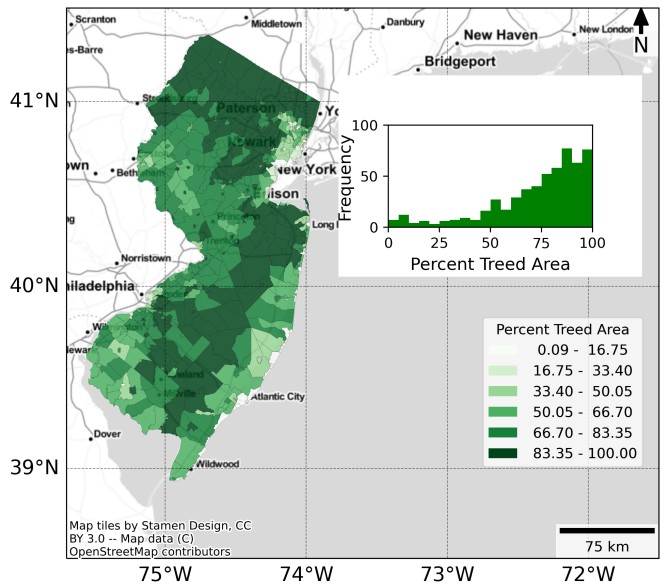

**Figure 4.** Distribution of percent tree area (Mean: 73.45%, Standard Deviation: 23.04%) across cities in New Jersey

## 2.7 Elevation

Previously, researchers have found that hurricane wind speeds (and thus damages) vary with surface topography (Chapman, 2000; Miller et al., 2013; Guikema et al., 2010; Quiring et al., 2011; McRoberts et al., 2018). Additionally, varying elevations could also introduce variations in precipitations (Napoli et al., 2019) and wind speeds (Chapman, 2000; Miller et al., 2013) within a city. Thus, we use the median and mean elevation at the city centroid, using nearest neighbor interpolation, as topographic variables to capture the changes in elevations across the cities. We obtained these parameters from the Digital Elevation Model at a 30m resolution scale developed by USGS as part of DEM: Global Multi-Resolution Terrain Elevation Data (GMTED2010) (Danielson and Gesh, 2011). Note that the resolution of available outage data at the city-level limits our

ability to account for the varying elevations within a city. Future studies with high-resolution outage data might account for the variations in elevation within a city.

## 2.8 Density Data

Demographics data is available from American Community Survey (ACS) (https://www.census.gov/programs-surveys/acs, last access: 21 September 2022). ACS collects different demographic data for each US census tract. ACS started data collection in 2010, and we have considered data from 2019. We obtained the population density as it indicates the number of distribution poles and system components exposed to winds (Liu et al., 2007).

## 3  Model Development: Feature Selection

Machine learning models with high dimensional input data can be hard to train, especially when datasets are sparse, as in the case of infrastructure failures. Input features can be correlated, leading to higher generalization errors. This means the machine learning model can fit well the training data, i.e., with small errors. However, we might observe significant errors after testing the model with additional data. Also, correlated features can lead to a flawed understanding of the relation between input and predicted outages (Verleysen and François, 2005).

Feature selection, also called variable selection, is an essential step in machine learning model development to select relevant variables and discard redundant and highly correlated ones (Cai et al., 2018). We performed the feature selection for outage prediction in two steps. First, we performed a forward selection with a linear regression (Kohavi and John, 1997) for an initial rank on feature importance (Figure 5). A linear model might not be the best model to forecast power outages. However, it can provide initial insights into the dependence of an input feature on outages. We started with a set of empty features and added features one by one. At each step, we selected the variable that led to the largest increase in the $R^2$. Our results show that, as expected, wind speed and duration of strong winds affect the power outages most. We found precipitation and soil moisture are important for outage prediction, even for linear regression, suggesting that their relevance could be even higher for non-linear regressions. We also found that population density is critical for outage prediction, which could be explained by a positive correlation between density and the density of transformers, as described in Liu et al. (2007).

| Feature | Abbreviation | Data Source | Previous Applications |
|---|---|---|---|
| Outages | Outages | poweroutage.us | - |
| 3-s Gust Wind Speed | Vmax* | Hurricane Parameters [1] | [9-18] |
| Duration of Strong Winds | Duration | | |
| Percent Developed Area | Developed | National Land Cover Data [2] | [9-18] |
| Percent Water Area | Water | | |
| Percent Barren Area | Barren | | |
| Percent Forest Area | Forest | | |
| Percent Scrub Area | Scrub | | |
| Percent Grassland Area | Grassland | | |
| Percent Pasture Area | Pasture | | |
| Percent Crops Cultivated Area | Crops | | |
| Percent Wetlands Area | Wetlands | | |
| Standard Precipitation Index 1 month | SPI1 | NLDAS2 [3,4] | [11-16] |
| Standard Precipitation Index 3 months | SPI3 | | |
| Standard Precipitation Index 6 months | SPI6* | | |
| Standard Precipitation Index 12 months | SPI12 | | |
| Soil Moisture $1^{st}$ Layer | CDF1* | NLDAS2 [3.4] | [11-16] |
| Soil Moisture $2^{nd}$ Layer | CDF2 | | |
| Soil Moisture $3^{rd}$ Layer | CDF3 | | |
| 7 day precipitation | Precip | NLDAS2 [3.4] | [9,10] |
| Root Zone Depth | Rzone* | Gridded Soil Survey [5] | [15] |
| Percent Treed Area | Trees* | NIDRM [6] | [15] |
| Mean Elevation | Mean_Ele | GTDEM 2010 [7] | [15,16] |
| Median Elevation | Median_Ele | | |
| Population Density | Pop_Den* | American Community Survey [8] | [14,15,16] |

**Table 1.** Parameters to build the power outage prediction models:all variables are rescaled at the city level. Parameters are grouped into categories separated by horizontal lines. We selected one variable from each category from each group to minimize correlation across parameters.

*-variables finally selected for model development after performing feature selection

Sources: [1] Chavas et al. (2015); [2] https://www.mrlc.gov/viewer/, last access: 21 September 2022; [3] Xia et al. (2012); Xia (2012); [5] Soil Survey Staff; [6] Krist Jr. et al. (2014); [7] Danielson and Gesh (2011); [8] https://www.census.gov/programs-surveys/acs, last access: 21 September 2022; [9] Liu et al. (2005); [10] Liu et al. (2008); [11] Han et al. (2009b); [12] Han et al. (2009a); [13]Nateghi et al. (2014); [14] Guikema et al. (2010); [15] McRoberts et al. (2018); [16] Shashaani et al. (2018); [17] Wanik et al. (2017); [18] Wanik et al. (2015)

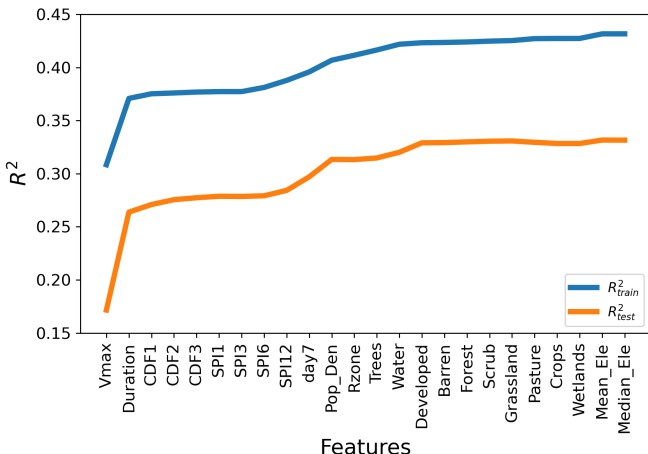

**Figure 5.** Forward Selection: Selection of important input parameters based on importance to explain the variability in outage predictions. Feature descriptions are shown in Table 1.

In the second stage, we analyzed the correlations between the input parameters. Supplementary Figure S4 shows the correlation coefficients for each pair of variables. We found that input features within the same category in Table 1 are highly correlated. For example, the maximum wind speed and duration of strong winds, which are at the top of the ranking in forward selection (Figure 5), has a correlation coefficient of 0.89 (Supplementary Figure S4). Hence, we kept only maximum wind speed as an input feature since it is better ranked than the duration of strong winds in the forward selection. We conducted a similar analysis for the different categories listed in Table 1 to select the input variable with the strongest predictive power. Due to their lower importance in our results, we did not include parameters from the elevation and land cover categories, as they contribute less than $1\%$ to $R^2$. Finally, we selected the following seven variables that will be used throughout the paper.

- 3-s Gust Wind Speed
- 7-day precipitation
- SPI 6 month
- Soil Moisture $1^{st}$ Layer
- Population Density
- Percent Treed Area
- Root Zone Depth

## 4 Generalized linear models

Generalized Linear Models (GLMs) are a generalization of ordinary linear regression. GLMs allow us to use a flexible link function to relate a linear model (of the input variables) to the response variable (Dunn and Smyth, 2018). Unlike ordinary

linear regressions, GLMs do not assumes homoscedasticity, *i.e.,* when the variance of the response variable is constant across the values of the input variables. The assumption of homoscedasticity fails for the number of customers without power since this output variable has positive counts, and when damage to power infrastructure is negligible (e.g., little storm), the variable's variance (and mean) should change and approach zero (Dunn and Smyth, 2018).

In addition, GLMs can utilize multiple statistical models to represent the data instead of the only normal distribution as in ordinary linear regressions. Outages have a lower bound of zero counts that normal distributions cannot capture. Thus, previous researchers have used the following distributions to represent outages with GLMs.

## 4.1 Poisson GLMs

Poisson regression models, a category of GLMs, are applicable for positive count data where observations are independent. Outages are modeled as a Poisson random variable:

$$P(y; \mu) = \frac{e^{-\mu} \mu^y}{y!} \tag{1}$$

where $y$ is the number of outages in a city. The Poisson distribution is described by the parameter $\mu$, the mean number of outages in a city. A log link connects the parameter $\mu$ to the input variables, which assures that $\mu$ is greater than zero.

$$ln(\mu) = \beta X \tag{2}$$

where $\beta$ is learned from the historical outages from extreme events, often through maximum likelihood estimation (MLE). MLE finds the value of $\beta$ that maximizes the probability of observing the data. Readers can refer to Dunn and Smyth (2018) for more information on MLE estimates for GLMs. We use *glm* package in R studio (https://www.rdocumentation.org/packages/stats/versions/3.6.2/topics/glm, last access: 21 September 2022) to fit the Poisson GLM to our power outage data.

The variance in a Poisson distribution is equal to the mean, i.e., $Var(y) = \mu$. Thus, the variance grows as $\mu$ increases. However, previous research has found that outage variance from historical data is significantly bigger than the mean (Liu et al., 2005; Han et al., 2009b, a), a phenomenon that is known as overdispersion in Poisson regressions (Dunn and Smyth, 2018). Overdispersion may arise from the interdependence of output variables, especially when they happen in clusters (Dunn and Smyth, 2018). Poisson distributions represent counts of events, e.g., customers without power, that are independent (Liu et al., 2007). In contrast, multiple outages in the city can happen due to the failure of the same power grid components (Liu et al., 2005; Han et al., 2009b). Thus, outage counts are not independent.

## 4.2 Negative Binomial GLMs

Negative Binomial GLM is a hierarchical model which can account for overdispersion effects in power outage count predictions (Dunn and Smyth, 2018). Negative Binomial GLMs are based on a Poisson-Gamma mixture distribution, i.e., the outage count $y$ is distributed as a Poisson random variable

$$P(y; \mu, \tau) = \frac{e^{-\mu\tau}(\mu\tau)^y}{y!} \tag{3}$$

where $\mu$ is a factor that when multiplied by $\tau$ equals the mean of the Poisson distribution. $\tau$ is an additional random variable to account for extra variance, with a mean equal to 1 and distributed as Gamma

$$P(\tau; k) = \frac{(1/k)^{1/k}}{\Gamma(1/k)} \tau^{1/k-1} e^{-\tau/k} \tag{4}$$

where $k$ is the overdispersion parameter and $\Gamma()$ is the gamma function, thus, the variance of $\tau$ equals $k$. After marginalizing the random variable $\tau$,

$$P(y; \mu, k) = \frac{\Gamma(y + 1/k)}{\Gamma(y+1)\Gamma(1/k)} \left( \frac{\mu}{\mu + 1/k} \right)^y \left( 1 - \frac{\mu}{\mu + 1/k} \right)^{1/k} \tag{5}$$

which is equivalent to a Negative Binomial distribution with a variance of $\mu + k\mu^2$. This variance is higher than the one in the Poisson GLM with one term that is proportional to $\mu^2$. Thus, Negative Binomial GLMs account for significantly higher variances. $\mu$ is parameterized as in Eq. 2. Then, $\beta$ and $k$ are estimated through through MLE, using the *glm* package in R studio ((https://www.rdocumentation.org/packages/stats/versions/3.6.2/topics/glm, last access: 21 September 2022)).

### 4.3 Zero-inflated GLMs

Researchers have also developed zero-inflation outage prediction models to improve statistical performance for unbalanced data, e.g., when there are a lot of data points with no outages (McRoberts et al., 2018; Shashaani et al., 2018). The zero-inflation model has two levels of predictions (McRoberts et al., 2018; Shashaani et al., 2018). The first level can be a logistic regression or a decision tree model to check if there is at least one power outage (Hall, 2000). The first level model predicts "0" in case of no outages and "1" in case of at least one outage. The second level is the regression model predicting the number of outages for cases where the prediction was "1" at the first level. In this paper, we do not fit zero-inflated models as our data is balanced, i.e., we observe at least one outage in each city.

## 5 Generalized additive models

GLM models assume a linear relationship between the logarithm of the mean number of outages and input parameters (Eq. 2). However, previous research has shown that they have non-linear relationships (Han et al., 2009a), which can be modeled with non-parametric extensions of GLMs (Yee, 2012). Generalized Additive Models (GAMs) capture non-linear relationships using smoothing functions

$$ln(\mu) = \beta_0 + \sum_{j=1} \beta_j f_j(x_j) \tag{6}$$

where $\mu$ is the mean outages for a city, $x_j \in X$ is the individual input parameter, $\beta_0$ is the intercept, and $f_j(x_j)$ are the smoothing functions for each input parameter. Some examples of smoothing functions are regression splines, B-splines, and P-splines. Splines of any order could be used to fit GAMs, but accuracy increases negligibly after the quartic degree (Yee, 2012). Thus, we used quartic order polynomials for all input variables, except for maximum wind speeds. For this variable, we reduced the order of the polynomial to 1 to always obtain a monotonically increasing relationship between winds and outages, as we would expect from the structural behavior of infrastructure against extreme loads. We used MLE to estimate GAMs' parameters through *iteratively reweighted least squares (IRLS)* (Wood, 2017) using the *MGCV* library in R studio.

Poisson GAM assumes the Poisson distribution on the number of outages as described in Eq. 1, with link function equal to Eq. 6. Similarly, negative Binomial GAMs assume the Poisson-Gamma distribution mixture for the number of outages as mentioned in Eq. 5, with link function equal to Eq. 6.

## 6 Random Forests

Random Forest regressions (Breiman, 2001) are non-parametric ensembles of decision trees that do not assume any underlying probability distribution for the decision variable. Tree-based methods are easy to build and powerful machine learning tools. Decision trees split at each node based on some criteria involving the value of a particular input variable. For regression trees, binary splits at each node are performed for each variable (Figure 6) (Hastie et al., 2001). The split with the largest reduction of squared errors is selected at each step. The splitting stops once there is no performance gain for the regression analysis. For the prediction, the decision tree will point to a final leaf node based on the criterion for the splitting of feature space, and the output for the decision tree is the average of the predicted variable,

$$\hat{y}_i = average(y_i | x_i \in L_m) \tag{7}$$

where $L_m$ is the final leaf node. (Fig. 6).

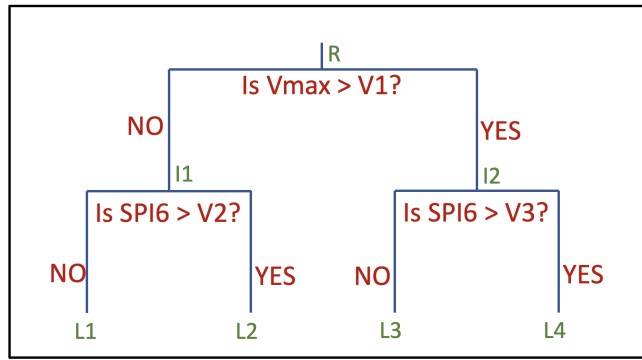

**Figure 6.** Example of a simple decision tree with two input features: maximum wind speed (Vmax) and precipitation (SPI6) to predict outages. Split at the root node ($R$) is done with Vmax. Thus, if the wind speed value is greater than the value $V1$, the points will belong to the right interior node ($I2$) otherwise they belong to the left node. Similarly, interior nodes are further divided into leaf nodes $L1, L2, L3$, and $L4$ using values of SPI6.

Random Forest "grow" a large number of parallel decision trees and bag new samples for each decision tree (Breiman, 2001). Bagging involves drawing new training points with replacements to fit each decision tree with a random selection of features. The final output is the average of outputs from each decision tree modeled in parallel. The random selection of features results in the development of uncorrelated trees, reducing the variance in predictions (Hastie et al., 2001).

Random Forest models can generally capture the non-linear between the input parameters and output predictions. However, a Random Forest is not easily interpretable as it is based on multiple decision trees. In this paper, we use the sci-kit learn module in python to fit the Random Forest model. We also use the GridSearchCV module in Python (Pedregosa et al., 2011) to tune for the parameters and select the model with the least error on out-of-bag samples.

## 7 Application of existing models

In this section, we discuss the statistical performance of different outage models by first training the models on training data and then comparing the $R^2$ metrics on hold-out test data. We use different $R^2$ metrics since traditional ones, like the coefficient of determination, have many limitations for counting variables, as discussed in the Appendix. We also compare the performance of developed models for our generalized data covering the Southeast, Southwest, and Northeast regions in the U.S. with the results from previous models applied to a particular region.

### 7.1 Generalized Linear Models

We trained Poisson and Negative Binomial GLMs to predict the outage counts. The predictions are based on the seven input features mentioned in the feature selection section. All the input features are significant at a p-value of $2 \times 10^{-6}$. We compared the statistical performance of the Poisson and Negative Binomial GLMs (Table 2). We have a total of 1,528 training data points with seven input variables and one additional slope constant. Thus, the residual degrees of freedom for each model is 1,520.

| Model | Residual Deviance | $R^2_{DEV}$ | $R^2_\psi$ |
|---|---|---|---|
| Poisson GLM | 6,038,042 | 0.20 | - |
| Negative Binomial GLM | 1,949 | 0.29 | 0.69 |

**Table 2.** Statistical performance measurements for Generalized Linear Models

The high value of residual deviance, relative to the degree of freedom, in the Poisson GLM, shows large overdispersion (Liu et al., 2005; Han et al., 2009b; Dunn and Smyth, 2018). Thus, using this new outage dataset, we confirm that the variance in historical outages largely exceeds the mean value. Negative Binomial GLM has a low residual deviance value compared to the Poisson model and is more similar to the degrees of freedom, indicating that Negative Binomial GLM can handle overdispersion in power outage predictions more satisfactorily.

$R^2_{DEV}$ in Table 2 is a measure of the deviance explained by the fitted model compared to the null model. The null model predicts the average of observed outages ($\bar{y}$) for all the cities irrespective of the input parameters. $R^2_\psi$ quantifies the amount of overdispersion explained by the additional variability introduced as a parameter in equations 4 and 5 for the Negative Binomial model fitted model. The $R^2_{DEV}$ is higher for the Negative Binomial GLM, also suggesting Negative Binomial's better statistical performance. The $R^2_\psi$ for the Negative Binomial GLM is $0.69$, which means the model can capture $69\%$ of variability by considering the additional level of uncertainty in the form of Poisson-Gamma mixture given by Eq.5 for outage counts. The reported value of $R^2_\psi$ for Negative Binomial GLM in this paper is comparable to the values presented by Han et al. (2009b), i.e., $\sim0.8$. However, we observe a lower value of $R^2_{DEV}$ compared to previous literature, i.e., $\sim0.6$. The lower value of $R^2_{DEV}$ may be due to the use of fewer parameters, e.g. seven in this study versus 20 in Han et al. (2009a), as $R^2_{DEV}$ always increases with more predictors. For example, we get a value of 0.48 for $R^2_{DEV}$ when all the input variables in Table 1 are included, but we considered fewer parameters to avoid correlated features and enhance the generalization of these models. We may also get a lower value of $R^2_{DEV}$ because we have a generalized dataset covering different regions in the U.S., and previous models were applied to data from smaller regions.

## 7.2 Generalized Additive Models

We also trained the Poisson and Negative Binomial GAMs. Like for GLMs, GAMs are trained with the seven input features mentioned before. All the input features are significant at a p-value of $2 \times 10^{-6}$ for both models. Like for GLMs, the residual degrees of freedom for each model is 1520.

| Model | Residual Deviance | $R^2_{DEV}$ | $R^2_\psi$ |
|---|---|---|---|
| Poisson GAM | 3,565,948 | 0.53 | - |
| Negative Binomial GAM | 1,866 | 0.62 | 0.99 |

**Table 3.** Statistical performance measurements for Generalized Additive Models

The residual deviance for Poisson GAM (Table 3) is ~38% less than the one in Poisson GLM (Table 2), which is a minimal improvement to overcome the large overdispersion as the deviance is still significantly higher than the degrees of freedom.

However, $R^2_{DEV}$ shows a more significant performance improvement as its value increases to 0.53 for Poisson GAM (Table 3) over a value of 0.20 for Poisson GLM (Table 2).

Negative Binomial GAM has a low value of residual deviance, which indicates that Negative Binomial GAM can handle overdispersion. Additionally, non-linear shapes from spline functions (Eq. 6) for GAMs improve the outage predictions. The $R^2_{DEV}$ for Negative Binomial GAM improves significantly to a value of 0.62 (Table 3) over the 0.29 in Negative Binomial

GLM (Table 2). The observed value of 0.99 for $R^2_\psi$ is similar to the one reported by Han et al. (2009a) in their GAM model development for power outage predictions. The $R^2_{DEV}$ for GAMs is not available in the previous literature, so further comparisons with previous work could not be made.

## 7.3 Random Forest

We calibrated the Random Forest model to predict the fraction of customers without power. We performed the hyperparameter

tuning using the GridSearch tool in Python (Pedregosa et al., 2011) with cross-validation to select the best input parameters for the Random Forest. The hyperparameter tuning resulted in a mean cross-validation $R^2$ of 0.52. However, we obtained a training $R^2$ of 0.84 when fitting the random forest with the tuned hyperparameters on the training data. The high training R2 compared to cross-validation $R^2$ represents potential overfitting in the Random Forest Model. We further tuned the model parameters by reducing the maximum depth. We obtained a cross-validation R2 of 0.48 and training $R^2$ of 0.63. Finally, we

obtained an $R^2$ of 0.48 on the hold-out test. The number of randomly grown trees in the selected Random Forest model is 500. Figure 7 shows the predicted fractional outages against the observed outages using a tuned RF. RF does not generalize well to the outages due to Hurricane Isaias, which had low wind speeds ($\sim 36 \ m/s$) in New Jersey, but still caused outages to $60\%$ of consumers in 213 cities (out of 565) in New Jersey. This effect is visible in Figure 7, where RF underestimated the fractional outages for severely affected cities in New Jersey.

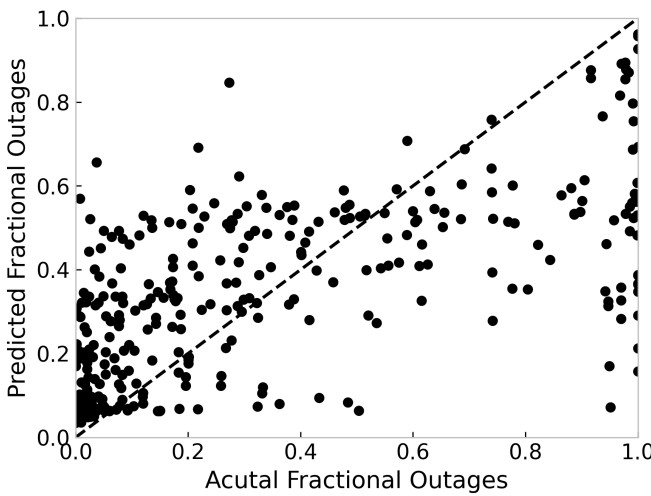

**Figure 7.** Random Forest outage predictions on the 20% holdout test data.

The $R^2$ for Random Forest model cannot be compared to different $R^2$ statistics calculated for GLM and GAM models as the output for Random Forest (fraction of outages) differs from GLM and GAM models (outage counts). Also, we cannot calculate $R^2_{DEV}$ and $R^2_\psi$ statistics for Random Forest as there is no underlying probability distribution assumed for Random Forest predictions.

We present the variable importance in the Random Forest model in Figure 8. We calculated the variable importance by training a base model with all the input features and a permuted model resulting from training different Random Forests and removing one feature at a time from the base model. We ranked importance by finding the variable that leads to the largest difference in the mean squared error between the base (full) model and the permuted (reduced) model. We present the normalized importance factors in decreasing order of importance (Figure 8).

We found that maximum wind speed is the most important parameter in the Random Forest model (Figure 8), which coincides with our findings from a simple linear regression in Figure 5. Precipitation is the second important variable, with a relative importance of 0.33, as trees can more easily be torn out from wetter soil. Population density is the third most important variable in outage predictions since it is a proxy for cities' density of transformers exposed to winds.

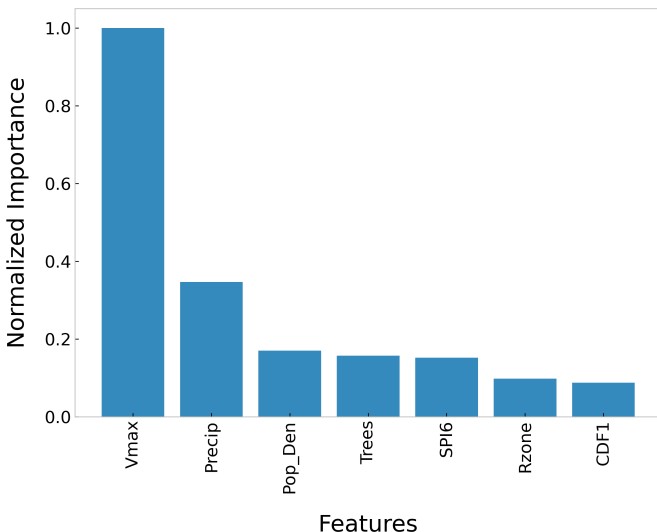

**Figure 8.** Random Forest variable importance in the decreasing order of importance. All importance factors are normalized by the highest value, i.e., the factor for Vmax.

Random Forest and Negative Binomial GAMs show superior performance in predicting the power outages caused by a hurricane. MSE (Wallach and Goffinet, 1989) have been used to compare the performance difference statistic models, given as.

$$MSE = \frac{1}{n}\sum(y - \hat{y})^2 \tag{8}$$

Where $y$ is the predicted value, $\hat{y}$ is the observed value, and $n$ is the total number of observations. Thus, we report MSE on both Negative Binomial GAM and RF models to compare the performance of these models. We rescale the Negative Binomial GAM predictions by the total number of customers to compare the MSE values of Negative Binomial GAM and RF models at the same scale as fractional outages. MSE for Negative Binomial GAM is 45.13, and MSE for RF is 0.06. Researchers should be careful in making the direct comparison for MSE values of the fraction-based RF model and the count-based Negative Binomial GAM model, as these models are optimized for a different set of response variables. The high MSE for the Negative Binomial arises from overestimating outages, which we discuss next.

## 8 Limitations of state-of-the-art outage models

Different machine learning models discussed in previous sections can predict power outages for a hurricane-stricken city. Here, we discuss the limitations of state-of-the-art machine learning models for power outage predictions.

## 8.1 GLM and GAM's Predictions are Unbounded

GLM and GAM regressions can predict the mean number of outages in the city. The models have a lower bound of zero as both Poisson, and Negative Binomial distributions predict the count of outages. However, there is no upper bound on the predicted number of outages. Hence, GLM or GAM models can predict more outages than the number of customers, resulting in an overestimation of power outages.

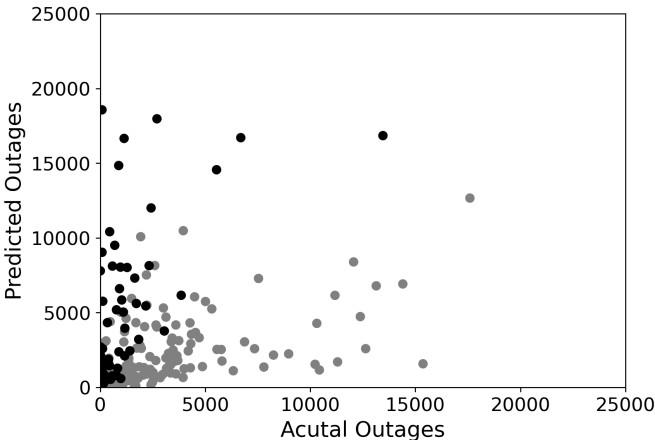

**Figure 9.** Outage predictions on 20% holdout test data using Negative Binomial GAM outage model. Black dots represent the cities with predicted outages larger than the number of customers. Grey dots represent the cities with predicted outages less than or equal to number of customers.

For illustration, we present the power outage predictions on 20% hold-out test data for the Negative Binomial GAM (Figure 9). For 76 cities out of 382 (19.8%) in test data, predicted outages are more than the number of customers, and the over-estimation can be significant. For example, the model predicted outages as high as 16 times that of Rockleigh, New Jersey customers. The average ratio of predicted outages over the number of customers in the cities that experience overestimation was 5.2. The cities that experienced overestimation had smaller populations, with an average of 5,962, e.g., Rockleigh had only 106 customers. In contrast, cities without overestimation had an average population of 30,058. Modelers could impose an upper bound on the predictions using the total number of customers as the maximum possible outage. However, this adjustment would violate the assumptions in the Poisson-Gamma mixture model (Eq. 5) and GAM link function (Eq. 6).

## 8.2 Random Forest's Lack of Extrapolability for High Winds

Random Forest predictions are an average value of the outages in the training data (Eq. 7). Thus, unlike GLMs, Random Forest predictions are bounded by the minimum and maximum values of power outages in training data. Based on simple physics, one would expect more damage and more outages from higher wind speeds. In order to understand the influence of wind speeds on the power outages in the Random Forest regression, we plotted the partial dependence of the fraction of customers without

power against wind speed. The partial dependence, $g(x_j)$ (Hastie et al., 2001; Nateghi et al., 2014) of the input variable $x_j$ is given by

$$g(x_j) = \frac{1}{N} \sum_{i=1}^{N} \hat{y}_i(x_j, x_{ic}) \tag{9}$$

where $N$ is the total number of observations, $x_{ic}$ are the variables other than $x_j$, and $\hat{y}_i$ is outage prediction (Eq. 7) for the $i^{th}$ data point. To plot the partial dependence, we varied $x_j$ (wind for this case) and kept $x_{ic}$ (input parameters other than wind speed) constant. We estimated outages by averaging all observations in training data plotted against the $x_j$ (wind speed).

For this assessment, we trained the Random Forest model on a reduced dataset with only New York and New Jersey and on a complete dataset, including Florida and Texas. We present the partial dependence of wind speed in Figure 10, also including the distribution of wind speeds in the training data. Hurricanes of category 3 or higher bring wind speeds above $40\,m/s$ that can significantly damage electric poles (Bjarnadottir et al., 2013). However, they are significantly less observed in inland cities, especially in the Northern United States, as storms often weaken in their transition to higher latitudes and after leaving the ocean. For example, only tropical storms with wind speeds of less than 33 m/s have impacted New York City for the past 20 years (https://coast.noaa.gov/hurricanes/#map, last access: 21 September 2022). For example, Superstorm Sandy (2012) transitioned to a tropical storm before impacting New York City (https://coast.noaa.gov/hurricanes/#map, last access: 21 September 2022). As per ASCE 7-10 wind hazard maps (https://hazards.atcouncil.org/, last access: 21 September 2022), a wind speed of $43\,m/s$ has a return period of 100 years for New York City. Thus, it is very unlikely and evident from Figure 10 to get outage data in New York and New Jersey for winds above $43\,m/s$.

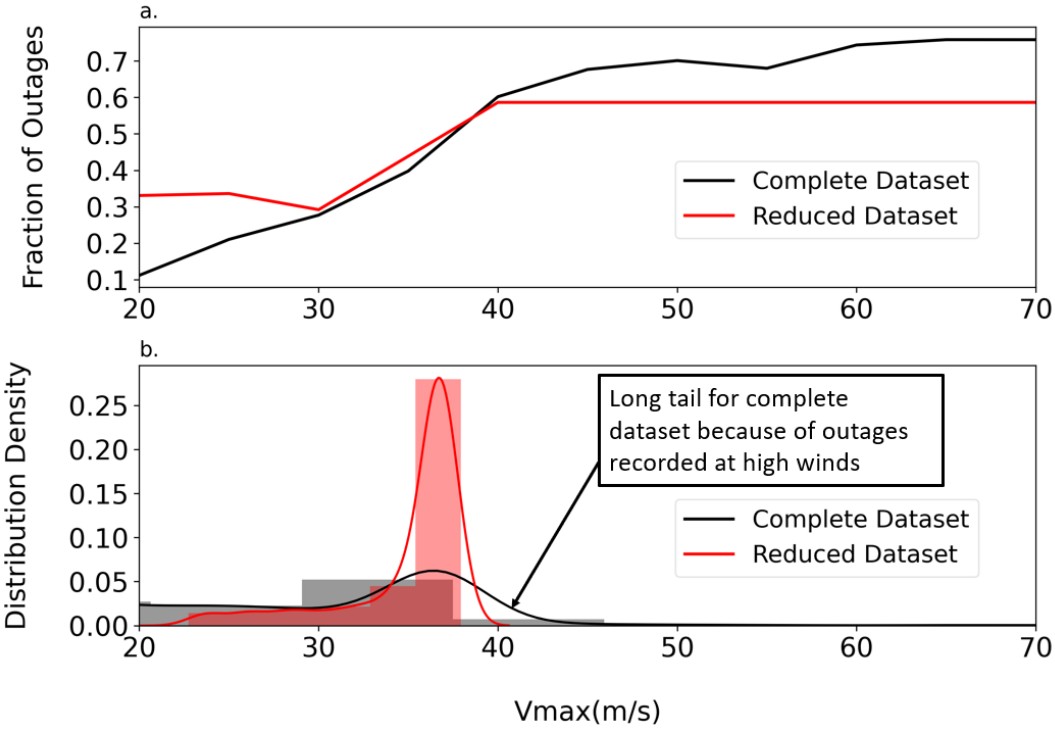

**Figure 10.** (a) Partial dependence of wind speed on power outages. (b) Distribution of wind speed in the complete and reduced dataset. Random Forest model does not extrapolate for the wind speeds and outages not in the range of training data.

These limited outage datasets have strong implications for the validity and extrapolability of outage models based on Random Forest regressions. Under the reduced dataset, i.e., with only New Jersey and New York, outage predictions increase as the wind speed increases from 20 to $40\,m/s$. However, the fraction of outages reached a maximum of 0.58 at wind speeds of $40\,m/s$ and does not increase with higher wind speeds. The Random Forest model cannot extrapolate for the higher winds, which limits the capability of Random Forest to make outage predictions for large hurricanes.

Under the complete dataset, results improve by including outages in Florida and Texas. These states experience higher winds, e.g., their 100-year return-period winds are $\sim 70\,m/s$ in contrast to the $43\,m/s$ in New York (https://hazards.atcouncil.org/, last access: 21 September 2022). The reduced dataset (with only New York and New Jersey) did not have any data points with wind speeds above $40\,m/s$. In contrast, the complete dataset (including Florida and Texas) had 88 cities (4.6% of data points) with winds greater than $40\,m/s$ and 29 cities (1.5% of data points) with winds above $70\,m/s$. With the complete dataset, the Random Forest predictions reach a maximum value of 0.76 for winds of $75\,m/s$. While these results show improvement, they also show that the data is still insufficient to make the Random Forest model follow the physics of infrastructure failure and extrapolate predictions for high winds causing outages close to 100%.

## 8.3 Lack of physics-based variance shapes

In catastrophic storms, we expect large outages with higher certainty, e.g., Hurricane Ida (2021) in Louisana(Elamrouss, 2021) and Tropical Storm Fiona (2022) in Puerto Rico (Rivera et al., 2022), close to 100%. Structural models for power poles estimate failure probabilities close to 95% for winds of $70\,m/s$ (Ouyang and Dueñas-Osorio (2014); Bjarnadottir et al. (2013)). Thus, the physics of infrastructure failure suggests that variance in outages should be smaller for catastrophic winds. To evaluate if existing models follow these principles from the physics of power infrastructure failure, we quantified the variance in predictions for Jersey City by varying the wind speed and keeping the other input variables unchanged (Figure 11).

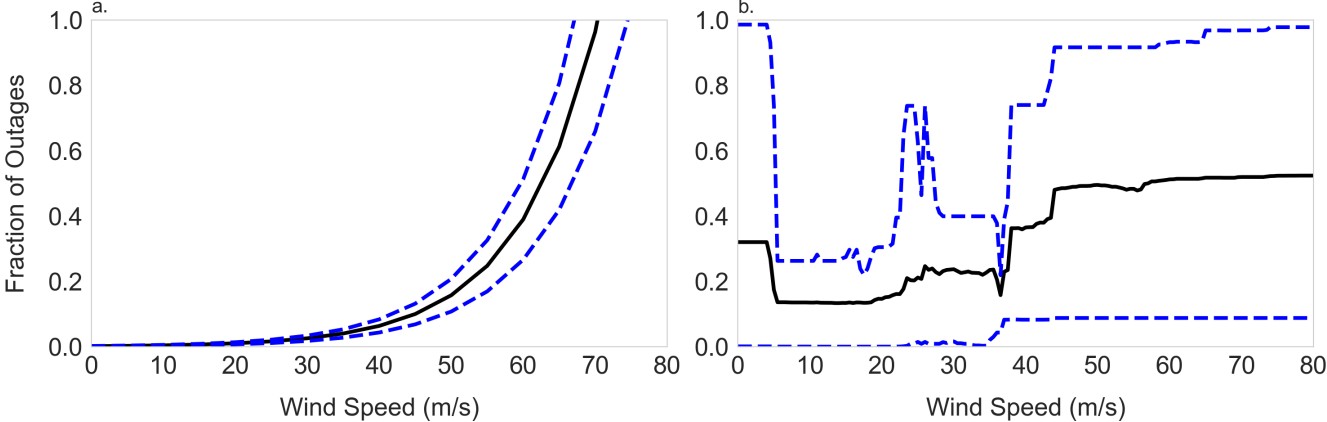

**Figure 11.** Prediction Ranges of Power Outage as a function of wind speed for (a) Negative Binomial GAM, (b) Random Forest. For GAM, mean and standard deviations are based on Eq. 5. For Random Forest, we use Quantile Random Forest to determine percentiles and assume Normal distribution to find comparable intervals of the mean plus and minus the standard deviation. Black lines indicate the mean outage predictions. Blue dashed lines indicate the one standard deviation interval for outage predictions.

As discussed previously, Negative Binomial GAMs capture the variability of outages better than Poisson models. Thus, we focus on the former models and estimate the variance according to Eq. 5. Figure 11a shows the mean and one standard deviation interval for outage predictions with varying wind speeds. We normalized the GAM's predictions to show fractions and compare them to the Random Forest model. GAM's predictions go beyond 1, as discussed previously, but we truncated the y-axis at 1 for comparison purposes. The linear relationship in the link function ensures that the variance (function of the square of mean outages 5) grows with wind speed. For example, for a wind of $40\,m/s$, we have a standard deviation of 0.02, and for a wind of $70\,m/s$, we have a standard deviation that is 15 times higher with a value of 0.3. Thus, the variance shows higher values as the predicted outage fraction approaches to 1. In fact, the variance is also unbounded in Negative Binomial GAM and goes to $\infty$.

Random Forest model can only predict the mean number of outages. Thus, it cannot evaluate variances. However, Quantile Regression Forest (QRF) (Meinshausen, 2006) can predict the outages at different confidence intervals, and we use it to quantify variance. The QRF uses the recorded observation at the leaf node to predict confidence intervals. These intervals are

fully data driven as Random Forests do not assume any underlying probability distribution on predicted outages (Ahsanullah et al., 2014). We presented Random Forest prediction intervals in Figure 11b. Random Forests had a standard deviation of 0.45 for high winds ($> 70\,m/s$), departing from the expected value of zero for catastrophic winds. Additional data could improve these Random Forest variance estimates. However, as mentioned earlier, infrastructure failure data is sparse.

Moreover, structural models predict no damage to power infrastructure at wind speeds lower than $10m/s$ (IEEE, 2007; Bjarnadottir et al., 2013). Thus, we expect outage predictions closer to 0 with a higher degree of certainty. Negative Binomial (and Poisson) GAMs handle this case well as the variance is zero when the mean outage is zero (Eq. 1 and 5). In contrast, we found that Random Forests had a standard deviation of 0.66 for zero wind speeds, showing that they also have limitations to represent physics-informed variance at low wind speeds.

As standard deviation is high for outages with the variation of winds and precipitation is the second most important variable in the RF model per Figure 8, we explore the relationship between outage fraction and precipitation to understand the non-linearity in outages. We have included the partial dependence plot of precipitation in Figure 12. We can observe from Figure 12, similar to 11, that precipitation also explains limited variability in the outages in this case. Also, at zero wind speed and zero precipitation RF model predicts non-zero outages. Next, we discuss the consequences of the limitations of the state-of-the-art power outage models.

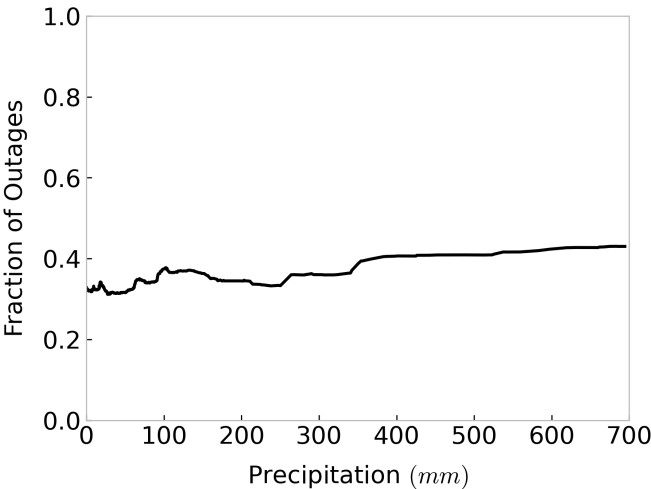

**Figure 12.** Partial dependence plot of fractional outages with precipitation.

## 9  Discussion

Utilities and governments often have limited resources to deploy for emergency response pre-event and during the event. Thus, they benefit from the power outage predictions ahead of a hurricane as it gives them a chance for effective and efficient pre-disaster planning. Jersey Central Power& Light (JCPL), a major utility company in New Jersey, serves 1.1 million consumers. Hurricane Isaias (2020) severely impacted JCPL's power supply, leaving 780,000 consumers without power in New Jersey

(Jim Giuliano, 2020). Hurricane Isaias impacted about 8,800 locations, with tree-related damages damaging 700 utility poles, 2,800 cross arms, 600 transformers, and around 80 miles of wire. JCPL mobilized around 1800 crew members to restore services. JCPL restored power for $86\%$ of the consumers in 72 hours (Jim Giuliano, 2020). Robust estimates of spatially distributed outages ahead of a hurricane can assist these utilities in planning their crews and potentially requesting additional ones from other utilities under mutual assistance agreements during disasters (National Academies of Sciences, Engineering, and Medicine, 2017).

Overestimated power outages could result in prioritizing a less affected city, placing more resources on that city than required. Limited mobility during a disaster can lead to prolonged outages, delaying the restoration effects (National Academies of Sciences, Engineering, and Medicine, 2017). In general, erroneous power outage estimates can result in the non-optimal placement of resources, as optimal resource allocation algorithms will use predicted outages (Brown, 2002). For example, JCPL predicted 449,312 customers without power ahead of Isaias compared to actual outages of 780,000 and mobilized only 1800 crew members restoring power outages for all customers after a week (Jim Giuliano, 2020) . The limited predictability of power outage models hinders the utilities' ability to arrange their crews and request help from other utilities to repair the power network effectively.

Additionally, large manufacturing companies or data centers are covered with business interruption insurance. Power outages from hurricanes can severely impact the operations of these companies. Insurance companies can use simulations on historical disaster data with power outage predictions to decide the insurance premium. Insurance premiums are based on downtime, the time for which power downtime. While calculating downtime is not the scope of this paper, the number of outages is determined to get the downtime (Liu et al., 2007). To determine downtime, insurance companies consider the uncertainty of the disruption, high uncertainty in predictions can lead to high insurance (Johnson, 2001). Thus, more certain estimates can help in more fair pricing of insurance premiums. Thus, improvements are required to make robust power outage predictions.

## 10 Suggested future research for improved outage modeling

Efforts are still needed to overcome the limitations of state-of-the-art power outage prediction models. In this paper, we suggest the study of Beta GAMs to address them (Olkin and Liu, 2003; Ferrari and Cribari-Neto, 2010). While performing a Beta regression analysis falls outside this paper's scope, we argue that the fundamental properties of Beta distributions can help overcome the challenges with state-of-the-art power outage prediction models. Beta distributions model random variables that take values from 0 to 1. Thus, Beta GAMs can model the fraction of outages in a city, making predictions of total outages that are bounded by the maximum number of customers, unlike Negative Binomial GLM and GAM regressions.

In addition, similar to Negative Binomial GAM, Beta GAM can account for the high variability of the input variables and can handle the overdispersion in power outage data (Douma and Weedon, 2019). Also, Beta GAMs could extrapolate outages for the extreme (low and high) values of winds since Beta predictions always go from their minimum value of 0 to their maximum value of 1. Finally, Beta GAMs have also closer to zero variance at observations values of zero and one. Thus, Beta GAMs

can also predict the outages with low variance at low and high winds closer to the physics of infrastructure failure. Thus, we suggest developing such Beta GAMs for improved outage prediction.

## 11    Conclusions

This paper summarized existing power outage prediction models; (a) GLMs and (b) GAMs based on Poisson and Negative Binomial distributions, and (b) Random Forest regressions. Power outages depend on several factors, including hurricane, environmental, and demographics conditions. To examine the existing models, we used power outage data with a total of 3.6 million outages for Hurricane Isaias (2020) in New York and New Jersey states, Hurricane Harvey (2017) in Texas, and Hurricane Michael (2018) in Florida. We combined the outages from these states to develop a generalized power outage model across different regions, improving previous efforts that only calibrated outage models to a particular region or utility companies in the U.S. We conducted a feature selection to avoid multi-collinearity among input variables and calibrated the state-of-art outage models using seven input parameters: 3-s wind gust speed, 7-day precipitation after the storm, standard precipitation index for 6 months before the storm, soil moisture for a depth between 0 and 10 cm, population density, the percent area covered by trees, and trees' root zone depth.

First, we found that Poisson regressions are unsuitable for modeling outages as historical outages have larger variances than the mean, resulting in overdispersion. The overdispersion was evident by the large residual deviances of 6,038,042 and 3,565,948 for the Poisson GLM and GAM, respectively, for 1520 degrees of freedom. We found that Negative Binomial regressions account for these larger variances better than Poisson regressions since we obtained residual deviances of 2,078 and 1,813 for GLM and GAM, respectively. We also showed that GAMs could better model the non-linear behavior of predictors compared to GLMs since $R^2_{DEV}$ and $R^2_{\psi}$ significantly increased to 0.62 and 0.99, respectively, compared to values of 0.29 and 0.69 in Negative Binomial GLMs. We demonstrated that the Random Forest could also capture this non-linear behavior as we found a value of 0.48 for the $R^2$ in the cross-validation.

However, each model has its own merits and demerits in predicting outages. Poisson and Negative Binomial estimates are unbounded and can overestimate power outages. For example, the Negative Binomial regression predicted more outages than the number of customers for $19.8\%$ of cities in the test data with an overprediction ratio of 5.2 for predicted outages compared to actual number of customers. Random Forest predictions are hard to calibrate for extreme winds as outage data is limited. As a result, we found that they could not be extrapolated for high winds since we only had 1.5% observations with wind speeds greater than $70\,m/s$. Negative Binomial GAM failed to account for small uncertainty in outage predictions at high winds as we observed that instead, the standard deviation in predictions grew 15 times with increasing wind speed from $40\,m/s$ to $70\,m/s$. We found that Random Forest also fails to account for small uncertainty at low winds.

Finally, we suggested that Beta regressions can address the shortcomings of existing power outage prediction models. The Beta regression outage prediction model can (1) make bounded outage predictions, (2) extrapolate for higher winds, and (3) model physics-informed variance in power outage predictions. Developing a Beta regression model will improve outage predictions to better guide utilities in placing their resources before and during the hurricane for a rapid recovery of the failed

power system. This improved regression model could also inform utilities about existing vulnerabilities of the power grid for
hardening.

*Code and data availability.* Applicable codes are available from author upon reasonable request. Authors can direct to third party for outage
data. Sources for publicly available datasets are included in this article.

## Appendix A

### A1 $R^2$ Parameter

$R^2$ parameter, a goodness-of-fit measure, is used to compare and select among different models. The goodness-of-fit can
quantify how good predictions are by the fitted model on unseen or test data.

$$R^2 = 1 - \frac{RSS}{TSS} \tag{A1}$$

The $R^2$ parameter mentioned in Eq. A1 represents the amount of variability explained by the fitted compared model to the
null model. The null model predicts the average of observed outages ($\bar{y}$) for all the cities irrespective of the input parameters.
In the Eq. A1, TSS is the residual sum of squares, defined by the sum of squares of the difference between the true value of the
response variable and average of true values of response variable.

$$TSS = \sum_i^N (y_i - \bar{y})^2 \tag{A2}$$

RSS is the total sum of squares, defined by the sum of squares of the difference between the true value of the response
variable and the predicted value of the response variable from the fitted model.

$$RSS = \sum_i^N (y_i - \hat{y})^2 \tag{A3}$$

### A2 $R^2_{DEV}$ Parameter

We quantify overdispersion by calculating if the residual deviance is larger than the degrees of freedom. Degrees of freedom is
defined as the number of data points in the training data minus the number of input parameters. We estimate

$$Deviance = -2(LL(sat) - LL(fit)) \tag{A4}$$

where $LL(sat)$ is the maximum achievable log-likelihood for the saturated model, and $LL(fit)$ is the log-likelihood for the
fitted model. Simplified versions of Eq. A4 to calculate deviance for different distributions are given below.

– Poisson: Residual deviance (Cameron and Windmeijer, 1996) for the $i^{th}$ observation for Poisson GLM and GAM is,

$$d_i(y_i, \hat{y}_i) = sign(y_i - \hat{y}_i) \cdot \left[ 2 \left\{ y_i log \left( \frac{y_i}{\hat{y}_i} \right) - (y_i - \hat{y}_i) \right\} \right]^{1/2} \tag{A5}$$

where $y_i$ are $\hat{y}_i$ are the observed and predicted outages for $i^{th}$ city. Deviance for the model is the sum of the square of the residual deviance for each observation.

$$D(y, y_i) = \sum_{i=1}^{N} (d_i(y_i, \hat{y}_i))^2; \tag{A6}$$

– Negative binomial: Residual deviance for the $i^{th}$ observation for a Negative Binomial GLM is

$$d_i(y_i, \hat{y}_i) = sign(y_i - \hat{y}_i) \cdot \left[ 2 \left\{ y_i log \left( \frac{y_i}{\hat{y}_i} \right) - (y_i + 1/k) ln \left[ \frac{y_i + 1/k}{\hat{y}_i + 1/k} \right] \right\} \right]^{1/2} \tag{A7}$$

and deviance for the fitted model is estimated by Eq. A6.

For GLMs and GAMs, a *pseudo-$R^2$*, denoted as $R^2_{DEV}$ (Cameron and Windmeijer, 1996), is also defined to compare the statistical performance based on model deviance. Similar to the definition $R^2$, $R^2_{DEV}$ measures the reduction in deviance of the fitted model when compared with the null model. The value of $R^2_{DEV}$ is given by,

$$R^2_{DEV} = 1 - \frac{D(y, y_i)}{D(y, \bar{y})} \tag{A8}$$

$D(y, y_i)$ is the deviance for the fitted model already defined in Eq. A4. For a null model, predictions will always be $\bar{y}$ (average of observed outages in training data). $D(y, \bar{y})$ is the deviance for a null model, which can be obtained by replacing $LL(fit)$ in Eq. A4 with $LL(null)$.

The value of $R^2_{DEV}$ will increase after adding more predictors as more predictors will always explain more variability in outage counts, decreasing the residual deviance. Also, the value of $R^2_{DEV}$ is bounded from 0 to 1, and a value closer to 1 will indicate a good fit of the model (Cameron and Windmeijer, 1996).

## A3   $R^2_k$ Parameter

$R^2_k$ is defined to measure the reduction in overdispersion for the fitted model Negative Binomial regression models when compared to the null model (Liu et al., 2005; Han et al., 2009b) .

$$R^2_k = 1 - \frac{k}{k_0} \tag{A9}$$

$k$ is the overdispersion factor (Eq. 5) for the fitted model, and $k_0$ is the overdispersion factor (Eq. 5) for the null model. Models with low overdispersion will have a low value of $k$. The, $R^2_k$ will be closer to one for a model with less overdispersion.

*Author contributions.* P.A. reviewed the existing models for power outage predictions during hurricanes, under L.C. 's supervision. P.A. and L.C. collected and curated the data for outages and input parameters, and fitted the power outages models for predicting outages. P.A. and L.C. conceptualized the Beta regression modeling framework for outages. P.A. drafted the manuscript with contributions and revisions from L.C.

*Competing interests.* The authors have no competing interests.

*Acknowledgements.* We acknowledge the financial support by the NYU Tandon School of Engineering Fellowship. Additionally, this research was also supported by the Coalition for Disaster Resilient Infrastructure Fellowship Grant 210924669. The authors are grateful for their generous support.

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
