# Peer review of "Probabilistic and Machine Learning Methods for Uncertainty Quantification in Power Outage Prediction due to Extreme Events"

_EGUsphere, 2022_

## Author Comment (AC1)

**Response to Reviewer#1**

This is a comprehensive work that compares different machine learning methods. I find the methods and presentations are solid and the authors did nice job in summarizing the substantial works they have finished. However, I do find some critical information is missing. **Namely, they need a comparison of performances for all ML models with the separate testing data because it will show whether their models have overfitting issues and how their performances for new data never encountered.** The author also needs more illustrations of how their model input data is obtained and possible associated uncertainties. Based on those, I suggest a major revision for this version. Please see detailed suggestions below.

Dear Reviewer,

We thank you for your review. We are grateful for your suggestions to help us get the manuscript in a better shape than the initial version. We have incorporated changes to the manuscript based on your insightful comments. We have included the error statistics on models, please see the response to comments 10 and 11. We have also included a discussion on associated uncertainties with the model input data, please see responses to comments 2, 3, and 5. Please find the detailed response to these and other specific comments.

**Comment#1:** Line 34, "including hurricane", is it wind field or else? Please clarify.

**Response#1:**

Thank you for pointing this out. Hurricane parameters used in the previous power outage studies and the current study are wind speeds and duration of strong winds. We have clarified more in L34 and modified the test to include "hurricane winds," so it would explicitly indicate using hurricane winds information for developing power outage models. We have modified the text to incorporate the changes shown below in bold.

Line 34: … These outage models use **inputs including hurricane winds**, power systems, environmental, and demographic information.

**Comment#2:** Uncertainty in the poweroutage.us data.

**Response#2:**

Potential uncertainties in the outage count can arise from the hindered communication with meters and utility monitoring points due to impacts from hurricanes. The utilities report the outages at the transmission level during the hurricanes as automatic reporting points are hindered (poweroutage.us/faq, last access: 13 January 2023). Another uncertainty in the outage data is the total customer count, as utilities do not report the total customers, and poweroutage.us calculates the total customers based on the maximum historical outage in a city (poweroutage.us/faq, last access: 13 January 2023). However, PowerOutage (poweroutage.us, last access: 21 September 2022) regularly gets updates from utilities to keep the outage count close to actual outages. Previous researchers obtained the data directly from utilities which are also not perfect (McRoberts et al., 2018; Shashaani et al., 2018) because the data had reported outages more than the number of customers. The more accurate reporting of outages and the

number of customers could improve the development of power outage predictions. To address your comment, we mentioned in lines 93-96 that power outage data recorded by poweroutage.us is obtained from utilities, so we assume outages reported by poweroutage.us are actual outages that happened during a hurricane. However, actual outages occurring in the field might be less or more than reported. Given the limited information about the outage management systems of individual utilities, we are not able to quantify uncertainty in outages for individual cities due to potential errors in the count of outages or total customers. For clarification, we have included in lines 93-96 these possible sources of uncertainties in outage data. The new text shown in bold reads as follows:

L93-96: We acquired power outage data from PowerOutage (poweroutage.us, last access: 21 September 2022), an organization that tracks and records outages from utilities at the city level across the U.S. **The automatic outage reporting points of utilities could be hindered during hurricanes which could result in errors in outage counts (poweroutage.us/faq, last access: 13 January 2023). However, PowerOutage (poweroutage.us, last access: 21 September 2022) regularly gets updates from utilities to keep the outage count close to actual outages.** The data covered the power outages for Hurricane 95 Isaias (2020) for 11 utilities in New York and 5 in New Jersey, for Hurricane Michael (2018) for 5 utilities in Florida, and for Hurricane Harvey (2017) for 39 utilities in Texas. Our dataset has about 3.6 million outages in total.

References

McRoberts, D. B., Quiring, S. M., & Guikema, S. D. (2018). Improving Hurricane Power Outage Prediction Models Through the Inclusion of Local Environmental Factors. *Risk Analysis*, *38*(12), 2722–2737. https://doi.org/10.1111/risa.12728

Shashaani, S., Guikema, S. D., Zhai, C., Pino, J. v., & Quiring, S. M. (2018). Multi-Stage Prediction for Zero-Inflated Hurricane Induced Power Outages. *IEEE Access*, *6*, 62432–62449. https://doi.org/10.1109/ACCESS.2018.2877078

**Comment#3:** Line 105 to 111, what are the possible uncertainties in interpolate all covariates into the city scale? Which interpolation method is used? Please be specific.

**Response#3:**

Thank you for bringing this up. We did not interpolate all the variables, so we have modified the "interpolation" to "methods" (used to obtain each parameter at the city scale) in line 109. These methods are later discussed in subsections (2.2-2.8) about each parameter. The modified text in lines 108 - 110 is as below:

(L108-110) **We obtained model inputs at the city level across all covariates. Further description of the availability, resolution, and methods to obtain each variable at the city level is provided in the subsequent subsections. We also discuss the uncertainties in data that could inherently influence the accuracy of power outage predictions.**

Following the reviewer's suggestions, we further discussed methods to obtain each parameter at the city scale and reviewed possible uncertainties that can arise in input parameters in subsections 2.2 - 2.8. Here, we present methods to obtain input parameters and uncertainties associated with these parameters:

**(Subsection: 2.2) Wind Speed**

We used a wind profile model for tropical cyclones (Chavas et al., 2015) to compute the wind speed at the city of the centroid. Hence, we do not perform any interpolation to obtain the wind speeds. Further, We investigated the variation in wind speeds across cities. We estimated the coefficient of variation (COV) of wind speeds for each city in the states of New Jersey and New York during hurricane Isaias (2020), Florida during hurricane Michael (2018), and Texas during hurricane Harvey (2017). We found that the COV for wind gusts is low, indicating wind gusts from the wind profile model vary little within a city. Thus, wind at the centroid of a city would be a reasonable estimate of wind gusts to determine the city-wide outages. We have incorporated the above uncertainty in wind gusts into the supplementary material. We have included supplementary section S1 with a discussion on uncertainty in wind gusts. The included section S1 reads as follows:

The wind speed is an important factor causing outages, and any uncertainty in wind speed estimates could lead to errors in outage predictions. We investigated the variation in wind speeds across cities. We estimated the coefficient of variation (COV) of wind speeds for each city in the states of New Jersey and New York during hurricane Isaias (2020), Florida during hurricane Michael (2018), and Texas during hurricane Harvey (2017). To compute COV, we first compute the 3-s wind gust at points spaced based on the tropical cyclone's wind profile (Chavas et al., 2015) at the 0.025º x 0.025º resolution in each state. Then, we compute the mean and standard deviation of wind gusts in each city by accounting for the points occurring within a city. Finally, the COV is computed as the ratio of standard deviation and mean. The high value of the COV can indicate the high variability in wind speeds. Figure S5 illustrates the 3-s wind gust in New Jersey during hurricane Isaias (2020) for points spaced at 0.025º  x 0.025º.

We studied how COV changes for different spacing of the points. The mean COV for wind gusts with 0.025º  x 0.025º points spacing in New Jersey is 0.001, New York is 0.003, Florida is 0.02, and Texas is 0.02. The change in COV for all states is minimal when increasing the resolution from 0.05º to 0.025º. Thus, given the minimal change in COV, we do not compute COV for resolutions higher than 0.025º. The low values of COV for wind gusts indicate that wind gusts at the centroid of a city would be a reasonable estimate of wind gusts to determine the city-wide outages. Figure S6 shows the histograms showing each state's COV distribution for wind gusts.

[Figure]

Figure S5. 3-s wind gust wind speeds for points spaced at 0.025º  x 0.025º

[Figure]

Figure S6. Distribution of Coefficient of Variation in 3-s wind gusts for (a) New Jersey (NJ) winds, (b) New York (NYS) winds, (c) Florida (FL) winds, (d) Texas (TX) winds

Accordingly, we have modified line 125 to incorporate the findings about the wind estimates. Now, we have included the visualizations showing the variability of a few parameters across multiple cities to give readers a context of the resolution of final input parameters and demonstrate the spatial scale of interest in power outage predictions.

Now, Line 125 is as follows (changes in bold):

… We determined the wind speed for each city at its centroid. **Figure 2 illustrates the variation of 3-s wind gusts in New Jersey during Hurricane Isaias. However, wind speed is an important factor causing outages, and any approximation in wind speed estimates could lead to errors in outage predictions. We provide detailed information in Supplementary Text S1 to demonstrate that wind speeds at the city centroid can be a reasonable estimate in determining the city-wide outages.**

[Figure]

Figure 2. Distribution of 3-s wind gust (Mean: 36.95 m/s, Standard Deviation: 0.41 m/s) across New Jersey before the arrival of Hurricane Isaias

**(Subsection: 2.3) Land Cover Data**

Land cover data is available at the city scale. We used zonal statistics in ArcGIS to obtain the percent area covered by each land cover type within a city. So, we do not interpolate for land cover data at the city scale. Hence, no interpolation errors or potential uncertainties are introduced when obtaining land cover at the city level. However, there may be uncertainties at the source of land cover data. National Land Cover Data has inaccuracies in the thematic pixel classification (Wickham et al., 2021) that could introduce uncertainty in the type of land cover (https://www.mrlc. gov/viewer/, last access: 21 September 2022). Addressing the source-level uncertainties is not within the scope of this paper. However, reduced inaccuracies in the future updates of input parameters could improve the outage predictions. We have included the statement on source-specific uncertainty in Lines 131-132 (changes in bold in subsection 2.3):

**Line 129:...** We obtained National Land Cover Data (NLCD) available from the Multi-Resolution 130 Land Characteristics Consortium, which is maintained by the United States Geographical Survey (USGS) (https://www.mrlc. gov/viewer/, last access: 21 September 2022). **National Land Cover Data has inaccuracies in the thematic pixel classification (Wickham et al., 2021) that could introduce uncertainties in the type of land cover. However, evaluating the effect of the inaccuracies in the thematic pixel classification on power outages is not within the scope of this paper**. NLCD data is available in raster format with a resolution of 30m x 30m. USGS has classified the original land cover data into 20 different classes. We have reclassified the NLCD data into nine to match previous power outage models. The nine different major classes of land cover data are developed area, water area, barren land, forest area, scrub area, grasslands, pasture land, cultivated cropland, and wetlands. We utilized the spatial analyst in ArcGIS (a 135 tool for Geographic Information Systems) (ESRI 2019) to clip the 30m x 30m land cover raster for each city. We used zonal analysis within ArcGIS to determine the percentage of area covered by the nine major land cover classes.

**(Subsection: 2.4) Precipitation and soil moisture**

We used nearest-neighbor interpolation to obtain soil moisture and precipitation at the city centroid by assigning values of these parameters available at the closest distance to the city's centroid. Soil moisture and precipitation are available at a resolution similar to the resolution of cities. Thus, interpolating these parameters at the city level would result in minimal variability. However, the limited temporal resolution of parameters required for computing soil moisture and precipitation could introduce errors in the final estimates of these variables (Wei et al., 2013). Addressing the limitations of the variable infiltration capacity (VIC) model from the National Land Data Assimilation System Phase 2 (NLDAS2) (Xia et al., 2012; Xia, 2012) to get soil moisture and precipitation is not the scope of this study. We have incorporated the changes to reflect the interpolation method (nearest neighbor) and source of uncertainty after Line 145 (changes shown in bold):

**Line 145:…**

Precipitation and soil moisture data are available from the variable infiltration capacity (VIC) model from National Land Data Assimilation System Phase 2 (NLDAS2) (Xia et al., 2012; Xia, 2012). **However, the limited temporal resolution of parameters required for computing soil moisture and precipitation could introduce errors in the final estimates of these variables (Wei et al., 2013). The limitations of the variable infiltration capacity (VIC) model from the National Land Data Assimilation System Phase 2 (NLDAS2) (Xia et al., 2012; Xia, 2012) to get soil moisture and precipitation are beyond the scope of this study.** Precipitation and soil moisture have been recorded each hour since 1979 with a resolution of $0.125° \times 0.125°$. **We used nearest-neighbor interpolation to obtain soil moisture and precipitation at the city centroid by taking the value available at the nearest point.**

Given the limited data within a city, we could not characterize the uncertainty for precipitation and soil moisture at this scale. However, we have now included the visualization showing the variability of the standard precipitation index at a 1-month scale across multiple cities. We have modified Lines 162-163 to include this figure (changes shows in bold):

 "**Figure 3 illustrates the variation of SPI 1 month in New Jersey before the arrival of Hurricane Isaias.**"

[Figure]

Figure 3. Distribution of SPI 1 month (Mean: 0.88, Standard Deviation: 0.93) across New Jersey before the arrival of Hurricane Isaias

**(Subsection: 2.5) Root Zone Depth**

Root zone depth is available at a higher resolution compared to the city scale. We averaged the root zone depth across the city to obtain the root zone depth as input at the city scale. Given the resolution of available outage data at the city scale, we were not able to consider the variations in root zone depth, which limits the ability to consider the variation of tree root strength within a city. We have mentioned the description of data in subsection 2.5 after Line 168 (changes in bold).

**Line 168:...** The effective root zone depth is defined as the depth of the soil from which plants and trees can effectively extract water and nutrients for growth (http://www.wood-database.com, last access: 21 September 2022). The more effective the root zone depth for trees, the less likely they will fail from strong hurricane winds (McRoberts et al., 2018). We add root zone depth as an input parameter for outage

predictions because it could indicate the hazard from falling trees to the power lines. Root zone data is available from the United States Department of Agriculture (USDA) under Gridded Soil Survey Geographic ((dataset) Soil Survey Staff) at 30m × 30m resolution as raster data. The root zone depth at the city level is calculated as the average of the root zone in a city using the spatial analytic tool ArcGIS. **Given the resolution of available outage data at the city scale, we were not able to consider the variations in root zone depth, which limits the ability of the power outage model to consider the variation of tree root strength within a city.**

**(Subsection: 2.6) Percent Tree Area**

Raster tree data has a higher resolution than a city. We do not need to interpolate for the area covered by the trees. Therefore, no errors or uncertainties arise from obtaining treed area at the city level. We have included the description of the dataset and visualization of variability in the percent treed area. We have done changes in the manuscript after line 176 (changes in bold):

**Line 176:…** USDA created National Insect and Disaster Risk Maps (Krist Jr. et al., 2014) in 2012, **with** the area covered by trees at 240m × 240m as raster data. The raster tree data is used to calculate the percent of the area covered by trees at the city level using the spatial analytic tool in ArcGIS (ESRI 2019). **Figure 4 illustrates the distribution of percent treed area in New Jersey.**

[Figure]

Figure 4. Distribution of percent treed area (Mean: 73.45%, Standard Deviation: 23.04%) across New Jersey

**(Subsection: 2.7) Elevation**

Similar to obtaining soil moisture and precipitation, we compute elevation at the city centroid using nearest neighbor interpolation to assign the value of elevation available at the nearest point to the city's centroid to capture the variability in elevation across cities. The varying elevation could introduce variations in precipitations (Napoli et al., 2019) and wind speeds (Chapman, 2000; Miller et al., 2013) within a city. The resolution of available outage data at the city-level limits our ability to account for the varying elevations within a city. Future studies with high-resolution outage data might account for the variations in elevation within a city. We have incorporated the statement on elevation data and corrected the resolution of elevation data after Line 180 (changes in bold):

**Line 180:...**

Previously, researchers have found that hurricane wind speeds (and thus damages) vary with surface topography (Chapman, 2000; Miller et al., 2013; Guikema et al., 2010; Quiring et al., 2011; McRoberts et

al., 2018). **The varying elevation could introduce variations in precipitations (Napoli et al., 2019) and wind speeds (Chapman, 2000; Miller et al., 2013) within a city. The resolution of available outage data at the city-level limits our ability to account for the varying elevations within a city. Future studies with high-resolution outage data might account for the variations in elevation within a city. For the scope of this paper, we use the median and mean elevation at the city centroid, using nearest neighbor interpolation, as topographic variables to capture the changes in elevations across the cities.** We obtained the mean and median elevation from the Digital Elevation Model at a 30-arc-second resolution scale developed by USGS as part of DEM: Global Multi-Resolution Terrain Elevation Data (GMTED2010) (Danielson and Gesh, 2011)

**(Subsection: 2.8) Density Information**

Density information is available at the city from American Community Survey (ACS). Hence, we do not interpolate for density information, so no within-the-city variation arises for obtaining the density information from ACS. However, Density information from ACS is based on a population sample, which introduces sampling error in determining the population (https://www.census.gov/programs-surveys/saipe/guidance/uncertainty.html, last access: 13 January 2023). Considering the sampling level error is outside the scope of this paper. We use the most recent estimates of ACS to get the most accurate available population estimates. Density information is included after Line 186.

Line 186:...Demographics data is available from American Community Survey (ACS) (https://www.census.gov/programs-surveys/acs, last access: 21 September 2022). ACS collects different demographic data for each US census tract. ACS started data collection, and we have considered data from 2019. We obtained the population density as it indicates the number of distribution poles and system components exposed to winds (Liu et al., 2007).

Reference:

Chapman, L. (2000). Assessing topographic exposure. *Meteorological Applications*, *7*(4), 335–340. https://doi.org/10.1017/S1350482700001729

Chavas, D. R., Lin, N., & Emanuel, K. (2015). A model for the complete radial structure of the tropical cyclone wind field. Part I: Comparison with observed structure. *Journal of the Atmospheric Sciences*, *72*(9), 3647–3662. https://doi.org/10.1175/JAS-D-15-0014.1

Danielson, J. J., & Gesh, D. B. (2011). *Global multi-resolution terrain elevation data 2010 (GMTED2010): U.S. Geological Survey Open-File Report 2011–1073*. Terrain. https://www.usgs.gov/publications/global-multi-resolution-terrain-elevation-data-2010-gmted2010

(dataset) Soil Survey Staff. (n.d.). *Gridded Soil Survey Geographic Database (gSSURGO) | Ag Data Commons*. Retrieved September 20, 2022, from https://data.nal.usda.gov/dataset/gridded-soil-survey-geographic-database-gssurgo

ESRI 2019. (n.d.). *ArcGIS Desktop: Release 10.8*.

Grimm, N. B., Faeth, S. H., Golubiewski, N. E., Redman, C. L., Wu, J., Bai, X., & Briggs, J. M. (2008). Global change and the ecology of cities. *Science*, *319*(5864), 756–760. https://doi.org/10.1126/SCIENCE.1150195/SUPPL_FILE/GRIMM.SOM.REV.PDF

Guikema, S. D., Quiring, S. M., & Han, S. R. (2010). Prestorm Estimation of Hurricane Damage to Electric Power Distribution Systems. *Risk Analysis*, *30*(12), 1744–1752. https://doi.org/10.1111/j.1539-6924.2010.01510.x

Krist Jr., F. J., Ellenwood, J. R., Woods, M. E., Mcmahan, A. J., Cowardin, J. P., Ryerson, D. E., Sapio, F. J., Zweifler, M. O., & Romero, S. A. (2014). *2013–2027 National Insect and Disease Forest Risk Assessment*. *January*.

Liu, H., Davidson, R. A., & Apanasovich, T. v. (2007). Statistical forecasting of electric power restoration times in hurricanes and ice storms. *IEEE Transactions on Power Systems*, *22*(4), 2270–2279. https://doi.org/10.1109/TPWRS.2007.907587

McRoberts, D. B., Quiring, S. M., & Guikema, S. D. (2018). Improving Hurricane Power Outage Prediction Models Through the Inclusion of Local Environmental Factors. *Risk Analysis*, *38*(12), 2722–2737. https://doi.org/10.1111/risa.12728

Miller, C., Gibbons, M., Beatty, K., & Boissonnade, A. (2013). Topographic speed-up effects and observed roof damage on Bermuda following Hurricane Fabian (2003). *Weather and Forecasting*, *28*(1), 159–174. https://doi.org/10.1175/WAF-D-12-00050.1

Napoli, A., Crespi, A., Ragone, F., Maugeri, M., & Pasquero, C. (2019). Variability of orographic enhancement of precipitation in the Alpine region. *Scientific Reports*, *9*(1). https://doi.org/10.1038/S41598-019-49974-5

Quiring, S. M., Zhu, L., & Guikema, S. D. (2011). Importance of soil and elevation characteristics for modeling hurricane-induced power outages. *Natural Hazards*, *58*(1), 365–390. https://doi.org/10.1007/s11069-010-9672-9

Wei, H., Xia, Y., Mitchell, K. E., & Ek, M. B. (2013). Improvement of the Noah land surface model for warm season processes: Evaluation of water and energy flux simulation. *Hydrological Processes*, *27*(2), 297–303. https://doi.org/10.1002/HYP.9214

Wickham, J., Stehman, S. v., Sorenson, D. G., Gass, L., & Dewitz, J. A. (2021). Thematic accuracy assessment of the NLCD 2016 land cover for the conterminous United States. *Remote Sensing of Environment*, *257*. https://doi.org/10.1016/J.RSE.2021.112357

Xia, Y. (2012). *NLDAS VIC Land Surface Model L4 Hourly 0.125 x 0.125 degree V002*. https://doi.org/10.5067/ELBDAPAKNGJ9

Xia, Y., Mitchell, K., Ek, M., Sheffield, J., Cosgrove, B., Wood, E., Luo, L., Alonge, C., Wei, H., Meng, J., Livneh, B., Lettenmaier, D., Koren, V., Duan, Q., Mo, K., Fan, Y., & Mocko, D. (2012). Continental-scale water and energy flux analysis and validation for the North American Land Data Assimilation System project phase 2 (NLDAS-2): 1. Intercomparison and application of model products. *Journal of Geophysical Research: Atmospheres*, *117*(D3), 3109. https://doi.org/10.1029/2011JD016048

**Comment#4:** Line 113: number of outages, is it the same as customers without power?

**Response#4:**

Yes, the number of outages is the same as customers without power. We have modified the text in Line 113 to describe this explicitly:

L113: **We focused on two response variables, the number of outages which is equivalent to the number of customers without power in a city, and the fraction of customers without power.**

**Comment#5:** Line 125: Uncertainty in wind speed estimates since the sizes of cities vary.

**Response#5:**

Thank you for pointing this out. Following the reviewer's suggestion, we investigated the variation in wind speeds across cities. We estimated the coefficient of variation (COV) of wind speeds for each city in the states of New Jersey and New York during hurricane Isaias (2020), Florida during hurricane Michael (2018), and Texas during hurricane Harvey (2017). The low values of COV for wind gusts indicate that wind gusts at the centroid of a city would be a reasonable estimate of wind gusts to determine the city-wide outages. We have explained the systematic study on COV of wind gusts in comment#3. Please see the detailed response to comment#3.

**Comment#6:** Line 147: Which rescaling technique is used for this one?

**Response#6:**

Thank you for pointing this out. We used nearest-neighbor interpolation to obtain soil moisture and precipitation at the city centroid by taking the value available at the nearest point. Instead, we computed the parameters at the city centroid by taking the value available at the point nearest to the city centroid. We have changed the *"rescaled"* to *"interpolation."* We have also mentioned this correction in comment#3. The rest of Line 147 reads as follows:

Line 147: … **"We used nearest-neighbor interpolation to obtain soil moisture and precipitation at the city centroid by taking the value available at the nearest point."**

**Comment#7:** Line 175: Please fix the citation.

**Response#7:**

Thanks for pointing this out. We will fix the citation.

**Comment#8:** Table 2: how to interpret the difference between R2DEV and R2ψ?

**Response#8:**

Thank you for your comment. We included the interpretations of $R^2_{DEV}$ and $R^2_{\psi}$ in the appendix of the preprint.

$R^2_{DEV}$ in Table 2 is a measure of the deviance explained by the fitted model compared to the null model. The null model predicts the average of observed outages $(\overline{y})$ for all the cities irrespective of the input parameters. $R^2_{\psi}$ explains the amount of overdispersion explained by the additional variability introduced as an overdispersion parameter (k) in equations 4 and 5 for the Negative Binomial model fitted model.

Now, we have included the simple interpretations after line 323:

"$R^2_{DEV}$ in Table 2 is a measure of the deviance explained by the fitted model compared to the null model. The null model predicts the average of observed outages ($\bar{y}$) for all the cities irrespective of the input parameters. $R^2_{\psi}$ explains the amount of overdispersion explained by the additional variability introduced as a parameter in equations 4 and 5 for the Negative Binomial model fitted model".

**Comment#9:** Why is random forest used only for the fraction of customers without power? Is the number of power outages not fit the RF algorithm?

**Response#9:**

Thank you for pointing this out. RF can predict both the number and fraction of outages. However, we assessed a fraction of outages in this paper to focus on the state of the art for outage prediction as it gives better results. (McRoberts et al., 2018) previously developed RF outage prediction models based on fractions with improved accuracy over the model predicting total outages, as fractional outages could be more useful than total outages because of high variability in the number of total consumers (Reilly & Guikema, 2015). For example, neighboring cities A and B have very similar input parameters (wind speeds, precipitation) and very different numbers of customers(say 1000 and 10000). However, cities A and B have almost the same fractional outages as both cities are hit relatively equally by hurricane winds. Suppose we train a model using the total number of customers with training data having more instances like city B in training. In that case, the model will overestimate the outages for city A (around 10000 outages for 1000 consumers). On the contrary, the power outage model with fractional outages will give similar results for cities A and B. We fitted the fraction of outages for a city with the Random Forest (RF) Model, which we have discussed in the manuscript. Just for testing purposes, we also fitted a model with the number of customers. However, we achieved better accuracy with fractional outages because of less variation in fractional outages compared to large variability in total outages, as already reported in (McRoberts et al., 2018).

Reference:

Reilly, A., & Guikema, S. (2015). Bayesian Multiscale Modeling of Spatial Infrastructure Performance Predictions with an Application to Electric Power Outage Forecasting. *Journal of Infrastructure Systems*, *21*(2), 04014036. https://doi.org/10.1061/(asce)is.1943-555x.0000222

McRoberts, D. B., Quiring, S. M., & Guikema, S. D. (2018). Improving Hurricane Power Outage Prediction Models Through the Inclusion of Local Environmental Factors. *Risk Analysis*, *38*(12), 2722–2737. https://doi.org/10.1111/risa.12728

**Comment#10:** Figure 5: what are R2 and other error statistics in the holdout test? It will be helpful to report them in the same figure.

**Response#10:**

Thank you for pointing this out. We have computed the mean squared error (MSE) (Wallach & Goffinet, 1989) on the holdout test for Negative Binomial Generalized Additive Model (GAM). The $R^2$ values are reported as a goodness-of-fit measure for linear models (Cameron & Windmeijer, 1996). (Cameron & Windmeijer, 1996) defined pseudo-$R^2$ ($R^2_{DEV}$ and $R^2_{\psi}$) to measure as a measure for goodness-of-fit for count data models, Poisson, and Negative Binomial regression models. Previously, researchers (Han, Guikema, & Quiring, 2009; Han, Guikema, Quiring, et al., 2009; Liu et al., 2007) have reported pseudo-$R^2$ for the data used to fit the model, which we reported in the original manuscript. MSE (Wallach & Goffinet, 1989) have been used to compare the performance difference statistic models, given as

$$MSE \ = \ \frac{1}{n}\Sigma(y \ - \ \widehat{y})^2$$

Where y is the observed value, $\widehat{y}$ is the predicted value, and n is the total number of observations

Thus, we report MSE on both Negative Binomial GAM and RF models to compare the performance of these models. We rescale the Negative Binomial GAM predictions by the total number of customers to compare the MSE values of Negative Binomial GAM and RF models at the same scale as fractional outages. We have included the MSE error on the holdout test and modified text from 365 to 369:

"MSE (Wallach & Goffinet, 1989) have been used to compare the performance difference statistic models, given as.

$$MSE \ = \ \frac{1}{n}\Sigma(y \ - \ \widehat{y})^2$$

Where y is the observed value, $\widehat{y}$ is the predicted value, and n is the total number of observations

Thus, we report MSE on both Negative Binomial GAM and RF models to compare the performance of these models. We rescale the Negative Binomial GAM predictions by the total number of customers to compare the MSE values of Negative Binomial GAM and RF models at the same scale as fractional outages.  MSE for Negative Binomial GAM is 8.19, and MSE for RF is 0.06. Researchers should be careful in making the direct comparison for MSE values of the fraction-based RF model and the count-based Negative Binomial GAM model, as these models are optimized for a different set of response variables.The high MSE for the Negative Binomial arises from the overestimation of outages which we discuss in section 8.1"

Reference:

Wallach, D., & Goffinet, B. (1989). Mean squared error of prediction as a criterion for evaluating and comparing system models. *Ecological Modelling*, *44*(3–4), 299–306. https://doi.org/10.1016/0304-3800(89)90035-5

**Comment#11:** Do you have any prediction vs. observation plot for the RF model like Figure 5?

**Response#11:**

Thank you for your suggestion. We have added a new figure 3 in the text representing the predicted fractional outages against the observed outages using Random Forest (RF).

As per your earlier suggestion, we kept a 20% holdout test data to check the out-of-sample accuracy of models and if any overfitting issues occur in the model. We performed the hyperparameter tuning using the GridSearch tool in Python (Pedregosa et al., 2011) with cross-validation to select the best input parameters for the RF model. We have modified the text in line 348. The new text in the 348 reads as follows:

"We performed the hyperparameter tuning using the GridSearch tool in Python (Pedregosa et al., 2011) with cross-validation to select the best input parameters for the Random Forest. The hyperparameter tuning resulted in a mean cross-validation $R^2$ of 0.52. However, we obtained a training $R^2$ of 0.84 when fitting the random forest with the tuned hyperparameters on the training data. The high training $R^2$ compared to cross-validation $R^2$ represents potential overfitting in the Random Forest Model. We further tuned the model parameters by reducing the maximum depth. We obtained a cross-validation $R^2$ of 0.48 and a training $R^2$ of 0.63. Finally, we obtained an $R^2$ of 0.48 on the hold-out test. The number of randomly grown trees in the selected Random Forest model is 500. Figure 3 shows the predicted fractional outages against the observed outages using a tuned RF. RF does not generalize well to the Isias, which had low wind speeds (~36 m/s) in New Jersey, but still caused outages to 60% of consumers in 213 cities (out of 565) in New Jersey. This effect is visible in figure 8, where RF underestimated the fractional outages for severely affected cities in New Jersey. Also, Jersey Central Power & Light, a power utility company in New Jersey, predicted a maximum of 449,312 customers without power, but 780,000 customers lost power during Hurricane Isaias (2020) (Giuliano, 2020). For future studies, more meteorological parameters related to hurricanes' pressure and translation speed could better capture the damages from Hurricanes."

[Figure]

Figure 8. Outage predictions on 20% holdout test with random forest regression.

Reference:

Giuliano, J. (2020). *State of New Jersey New Jersey Board of Public Utilities Review and Assessment of Electric Utility Performance August 4, 2020 Tropical Storm Isaias Weather Event Division of Reliability and Security*.

Pedregosa, F., Varoquaux, G., Gramfort, A., Michel, V., Thirion, B., Grisel, O., Blondel, M., Prettenhofer, P., Weiss, R., Dubourg, V., Vanderplas, J., Passos, A., Cournapeau, D., Brucher, M., Perrot, M., & Duchesnay, E. (2011). Scikit-learn: Machine Learning in {P}ython. *Journal of Machine Learning Research*, *12*, 2825–2830.

**Comment#12:** Section 8.2, there is a heavy discussion on how winds control the power outage from the models. However, how precipitation is related to power outages is not shown, as it is the second most important variable in the RF model. You have demonstrated some nonlinear relationships between wind speed and power outage fraction. Therefore, it is worthwhile to show precipitation's relationship to outrage fraction or show precipitation and wind jointly with outage prediction in a separate pdp plot. That may explain some nonlinear relationships in Figure 7b.

**Response#12:**

Thank you for your suggestion. We agree on exploring the relationship between outage fraction with precipitation, as precipitation is the second most important variable in the RF model per figure 4 in the initial manuscript. We have included the partial dependence plot of precipitation in figure 12. We can observe from figure 12, similar to figure 7b in the original manuscript, precipitation also explains limited variability in the outages. In the original manuscript, we mentioned that we expect outage predictions closer to 0 at lower winds, see lines 442-444:

"Moreover, structural models predict no damage to power infrastructure at wind speeds lower than 10m/s (IEEE, 2007; Bjarnadottir et al., 2013). Thus, we expect outage predictions closer to 0 with a higher degree of certainty."

However, at zero wind speed and zero predictions RF model predicts non-zero showing the limitation of representing physics-based predictions. We have included the extended discussion in the manuscript after line 446 as follows:

"We explore the relationship between outage fraction and precipitation, as precipitation is the second most important variable in the RF model per figure 4. The relationship  We have included the partial dependence plot of precipitation as supplementary figure 12. We can observe from figure S7, similar to figure 7b, that precipitation also explains limited variability in the outages. Also, at zero wind speed and zero predictions RF model predicts non-zero outages."

[Figure]

Figure 12. Partial dependence plot of fractional outages with precipitation.

Reference:

Bjarnadottir, S., Li, Y., & Stewart, M. G. (2013). Hurricane Risk Assessment of Power Distribution Poles Considering Impacts of a Changing Climate. *Journal of Infrastructure Systems*, *19*(1), 12–24. https://doi.org/10.1061/(asce)is.1943-555x.0000108

*National Electrical Safety Code, ANSI/IEEE Standard C2-2007*. (2007). *552*.

**Comment#13:** Section 9, the author mentioned beta regression may have better performance. But no comparison is made with the previous method. I suggest shortening the arguments after line 454 because there is no evidence in the paper supporting them.

**Response#13:**

Thank you for the suggestion. While beta regression analysis was not the scope of this paper, we wanted to suggest possible candidate methods to overcome the challenges with state-of-the-art power outage prediction models. The authors are currently exploring the possibility of beta regression as a power outage prediction model. Accordingly, we have modified section 9 on "Suggested future research for "comprehensive" outage risk assessments" to point out that Beta distributions are just a suggestion (and we are currently working on that) as follows:

Efforts are still needed to overcome the limitations of state-of-the-art power outage prediction models. In this paper, we suggest the study of Beta GAMs to address them (Ferrari and Cribari-Neto, 2010; Olkin and Liu, 2003). While beta regression analysis was not the scope of this paper, we suggest possible candidate methods to overcome the challenges with state-of-the-art power outage prediction models. The authors are currently exploring the possibility of beta regression as a power outage prediction model. Beta distributions model random variables that take values from 0 to 1. Thus, it can model the fraction of outages in a city. For illustration, we present the possible prediction ranges of outages with wind speed for a Beta distributed fraction of outages (Yee, 2012; Olkin and Liu, 2003; Douma and Weedon, 2019). Similar to Negative Binomial GAM, Beta GAM can account for the high variability of the input variables and can handle the overdispersion in power outage data (Douma and Weedon, 2019).

Beta GAMs can make bounded predictions on the percentage of customers without electricity. Beta GAMs could extrapolate outages for the extreme (low and high) values of winds. Beta predictions always go from their minimum value of 0 to their maximum value of 1. Beta GAMs could predict the outages with low variance at low and high winds closer to the physics of infrastructure failure. Thus, future research could focus on developing such Beta GAMs for outage prediction.

Reference:

Ferrari, S. L. P., & Cribari-Neto, F. (2010). Beta Regression for Modelling Rates and Proportions. *Http://Dx.Doi.Org/10.1080/0266476042000214501*, *31*(7), 799–815. https://doi.org/10.1080/0266476042000214501

Olkin, I., & Liu, R. (2003). A bivariate beta distribution. *Statistics and Probability Letters*, *62*(4), 407–412. https://doi.org/10.1016/S0167-7152(03)00048-8

Yee, T. W. (2012). *Package "VGAM" (Vector generalized linear and additive models)*. http://www.springer.com/series/692

Douma, J. C., & Weedon, J. T. (2019). Analysing continuous proportions in ecology and evolution: A practical introduction to beta and Dirichlet regression. *Methods in Ecology and Evolution*, *10*(9), 1412–1430. https://doi.org/10.1111/2041-210X.13234

**Comment#14:** Line 469 to 470, unlike linear models, RF does not have the assumption of non-collinearity.

**Response#14:**

Thanks for pointing this out. We agree that RF, GLM, and GAM do not have the assumption of non-collinearity. We use feature selection to use only important variables and develop generalized models with the least number of features to apply the models to all coastal cities of the United States. We have modified Line 469 to 470 as follows:

"We conducted a feature selection to use important input variables and calibrated the state-of-art outage models using seven input parameters…"

---

## Author Comment (AC2)

**Response to Reviewer#2**

This paper investigated the limitations of existing power outage models, including bounded prediction, out-of-distribution prediction, and physics-aware uncertainties The authors found some of the existing state-of-the-art models may generate unrealistic predictions, and cannot generalize well to extreme events that are not sufficiently represented in the training datasets. The authors discuss some potential ways to address the shortcomings of these models. I have some major comments that authors need to address before publication:

Dear Reviewer,

Thank you for reviewing our manuscript. We are grateful for your comments that have helped to better explain the critical aspects of power outage predictions. Please find detailed responses to your comments.

**Comment#1:** The problems mentioned by the authors, including limited generalization ability, unbounded predictions, and unreasonable uncertainty variations, are common problem for general machine learning models. Many machine learning community researchers proposed different methods to address these problems. How unique and critical are they for power outage predictions?

**Response#1:**

We agree with you that paper lacked a discussion on how critical limitations are with the state-of-the-art power outage predictions. We have included a new section 8 as "Discussion" in the manuscript:

"Utilities and government agencies benefit from the power outage predictions ahead of a hurricane as it gives them a chance for effective and efficient pre-disaster planning. However, utilities and governments often have limited resources to deploy for emergency response pre-event and during the event. Jersey Central Power & Light (JCPL), a major utility company in New Jersey, serves 1.1 million consumers. Hurricane Isaias (2020) severely impacted JCPL's power supply, leaving 780,000 consumers without power in New Jersey (Giuliano, 2020). Hurricane Isaias impacted about 8,800 locations, with tree-related damages damaging 700 utility poles, 2,800 cross arms, 600 transformers, and around 80 miles of wire. JCPL mobilized around 1800 crew members to restore services. JCPL restored power for 86% of the consumers in 72 hours (Giuliano, 2020). Robust estimates of spatially distributed outages ahead of a hurricane can assist utilities in asking for crews from other utilities under mutual assistance during disasters ("Enhancing the Resilience of the Nation's Electricity System," 2017).

Erroneous power outage estimates can result in the non-optimal placement of resources, as optimal resource allocation algorithms will use predicted outages (Brown, 2002). Overestimated power outages could result in prioritizing a less affected city, placing more resources on that city than required. Limited mobility during a disaster can lead to prolonged outages, delaying the restoration effects ("Enhancing the Resilience of the Nation's Electricity System," 2017). JCPL predicted 449,312 customers without power ahead of Isaias compared to actual outages of 780,000 (Giuliano, 2020). Limited generalization of the power outage model limits the utilities to arrange the correct number of crews under mutual assistance.

Large manufacturing companies or data centers are covered with business interruption insurance. Power outages from hurricanes can severely impact the operations of these companies. Insurance companies can

use simulations on historical disaster data with power outage predictions to decide the insurance premium. Insurance premiums are based on downtime, the time for which power downtime. While calculating downtime is not the scope of this paper, the number of outages is determined to get the downtime (Liu et al., 2007). To determine downtime, insurance companies consider the uncertainty of the disruption, high uncertainty in predictions can lead to high insurance (Johnson, 2001). Thus, more certain estimates can help in a more fair pricing of insurance premiums. Thus, improvements are required to make robust power outage predictions."

Reference:

Brown, R. E. (2002). *Electric Power Distribution Reliability*. http://www.dekker.com

Enhancing the Resilience of the Nation's Electricity System. (2017). In *Enhancing the Resilience of the Nation's Electricity System*. https://doi.org/10.17226/24836

Giuliano, J. (2020). *State of New Jersey New Jersey Board of Public Utilities Review and Assessment of Electric Utility Performance August 4, 2020 Tropical Storm Isaias Weather Event Division of Reliability and Security*.

Johnson, S. G. (2001). *JOURNAL OF INSURANCE COVERAGE Insurance Coverage for Power Outage Losses*.

Liu, H., Davidson, R. A., & Apanasovich, T. v. (2007). Statistical forecasting of electric power restoration times in hurricanes and ice storms. *IEEE Transactions on Power Systems*, *22*(4), 2270–2279. https://doi.org/10.1109/TPWRS.2007.907587

**Comment#2:** Now there is a variety of more complex power outage prediction models [1], are there any specific reasons for the authors to choose to evaluate traditional machine learning models? These traditional models are known to be less representative.

[1]Xie, Jian, Inalvis Alvarez-Fernandez, and Wei Sun. "A review of machine learning applications in power system resilience." In 2020 IEEE Power & Energy Society General Meeting (PESGM), pp. 1-5. IEEE, 2020.

**Response#2:**

Thanks for pointing this out. We have included an introduction to more complex power outage prediction models, namely neural networks, kernel methods such as support vector machines, and other tree-ensemble methods, such as AdaBoost, which can model non-linear relationships between input parameters and outages (Xie et al., 2020). However, data availability limits the applicability of these methods at a large scale for power outage predictions from extreme events.

Power Outage Models by (Liu et al., 2007; Han et al., 2009; Guikema et al., 2014; McRoberts et al., 2018; Shashaani et al., 2018) provide outage predictions at a coarser level compared to predictions at component. However, these models are mostly based on open-source, publicly available data and can be generalized at a larger scale to the coastal cities in the United States. Hurricane-caused outages are mostly

at the transmission level, which is responsible for city-wide outages (Brown, 2002; poweroutage.us/faq, last accessed: 13 January 2023) rather than the customer meter level. So, predicting city-wide outages can still guide utilities to arrange for crews and emergency backup power ahead of a storm. We have also included this discussion (in blue below) in the introduction section after Line 76:

Previously, researchers have used more complex power outage prediction models, namely neural networks, kernel methods such as support vector machines, and other tree-ensemble methods, such as AdaBoost, which can model non-linear relationships between input parameters and outages (Xie et al., 2020). (Kankanala et al., 2014) employed AdaBoost to predict weather-related power outages. (Kankanala et al., 2014) trained a separate model for each city for daily use, and they did not cover extreme weather outages. (Eskandarpour & Khodaei, 2018 and Eskandarpour et al., 2018) used power grid component-level data with support vector machines. (Rudin et al., 2012) ranked the power grid components (feeder failures, cables, joints, terminators, and transformers) based on their vulnerability to extreme weather events. (Haseltine & Eman, 2017) used a neural network to predict the failure of the power grid components for pre-storm. Such models will require specialized high-resolution power grid component-level data for each city which is not accessible given the data protocols of utility companies. (Sun et al., 2016) used Twitter (https://twitter.com; last accessed: 13 January 2023) data to predict real-time outages. (Jaech et al., 2018) used repair logs data employing Natural Processing with a Recurrent Neural Network to predict real-time outage durations. However, tweets (https://twitter.com; last accessed: 13 January 2023) and repair logs are available after the hurricane made an impact on the city. Thus, leveraging repair logs is not possible to predict outages for pre-event planning ahead of a storm. Hence, data availability limits the applicability of these methods at a large scale for power outage predictions from extreme events.

GLM (Liu et al., 2007), GAM (Han et al., 2009), and Random Forest based power outage prediction models (Guikema et al., 2014; McRoberts et al., 2018; Shashaani et al., 2018) provide outage predictions at a coarser level compared to predictions at component. However, these models are mostly based on open-source, publicly available data and can be generalized at a larger scale to the coastal cities in the United States. Hurricane-caused outages are mostly at the transmission level, which is responsible for city-wide outages (Brown, 2002; poweroutage.us/faq, last accessed: 13 January 2023) rather than the customer meter level. So, predicting city-wide outages can still guide utilities to arrange for crews and emergency backup power ahead of a storm. Hence for the scope of this paper, we focus on GLM, GAM, and Random Forest based power outage prediction models.

Reference:

Brown, R. E. (2002). *Electric Power Distribution Reliability*. http://www.dekker.com

Eskandarpour, R., & Khodaei, A. (2018). Leveraging accuracy-uncertainty tradeoff in SVM to achieve highly accurate outage predictions. *IEEE Transactions on Power Systems*, *33*(1), 1139–1141. https://doi.org/10.1109/TPWRS.2017.2759061

Eskandarpour, R., Khodaei, A., Paaso, A., & Abdullah, N. M. (2018). *Artificial Intelligence Assisted Power Grid Hardening in Response to Extreme Weather Events*. https://doi.org/10.48550/arxiv.1810.02866

Guikema, S. D., Nateghi, R., Quiring, S. M., Staid, A., Reilly, A. C., & Gao, M. (2014). Predicting Hurricane Power Outages to Support Storm Response Planning. *IEEE Access*, *2*(September 2015), 1364–1373. https://doi.org/10.1109/ACCESS.2014.2365716

Han, S. R., Guikema, S. D., & Quiring, S. M. (2009). Improving the predictive accuracy of hurricane power outage forecasts using generalized additive models. *Risk Analysis*, *29*(10), 1443–1453. https://doi.org/10.1111/j.1539-6924.2009.01280.x

Haseltine, C., & Eman, E. E. S. (2017). Prediction of power grid failure using neural network learning. *Proceedings - 16th IEEE International Conference on Machine Learning and Applications, ICMLA 2017*, *2017-December*, 505–510. https://doi.org/10.1109/ICMLA.2017.0-111

Jaech, A., Zhang, B., Ostendorf, M., & Kirschen, D. S. (2018). *Real-Time Prediction of the Duration of Distribution System Outages*. http://arxiv.org/abs/1804.01189

Kankanala, P., Das, S., & Pahwa, A. (2014). Adaboost+: An ensemble learning approach for estimating weather-related outages in distribution systems. *IEEE Transactions on Power Systems*, *29*(1), 359–367. https://doi.org/10.1109/TPWRS.2013.2281137

Liu, H., Davidson, R. A., & Apanasovich, T. v. (2007). Statistical forecasting of electric power restoration times in hurricanes and ice storms. *IEEE Transactions on Power Systems*, *22*(4), 2270–2279. https://doi.org/10.1109/TPWRS.2007.907587

McRoberts, D. B., Quiring, S. M., & Guikema, S. D. (2018). Improving Hurricane Power Outage Prediction Models Through the Inclusion of Local Environmental Factors. *Risk Analysis*, *38*(12), 2722–2737. https://doi.org/10.1111/risa.12728

Rudin, C., Waltz, D., Anderson, R., Boulanger, A., Salleb-Aouissi, A., Chow, M., Dutta, H., Gross, P., Huang, B., Ierome, S., Isaac, D. F., Kressner, A., Passonneau, R. J., Radeva, A., & Wu, L. (2012). Machine learning for the New York City power grid. *IEEE Transactions on Pattern Analysis and Machine Intelligence*, *34*(2), 328–345. https://doi.org/10.1109/TPAMI.2011.108

Shashaani, S., Guikema, S. D., Zhai, C., Pino, J. v., & Quiring, S. M. (2018). Multi-Stage Prediction for Zero-Inflated Hurricane Induced Power Outages. *IEEE Access*, *6*, 62432–62449. https://doi.org/10.1109/ACCESS.2018.2877078

Sun, H., Wang, Z., Wang, J., Huang, Z., Carrington, N., & Liao, J. (2016). Data-Driven Power Outage Detection by Social Sensors. *IEEE Transactions on Smart Grid*, *7*(5), 2516–2524. https://doi.org/10.1109/TSG.2016.2546181

Xie, J., Alvarez-Fernandez, I., & Sun, W. (2020). A review of machine learning applications in power system resilience. *IEEE Power and Energy Society General Meeting*, *2020-August*. https://doi.org/10.1109/PESGM41954.2020.9282137

**Comment#3:** It is unclear to me why beta regression should perform well in general cases. I think it also has its own problems such as strict distribution assumption, and does not address the representativeness issues which eventually cause the poor generalization problem. Could you provide any justifications and performance comparison regarding why Beta regression should be used?

**Response#3:**

Thank you for pointing this out. While beta regression analysis was not the scope of this paper, we wanted to suggest possible candidate methods to overcome the challenges with state-of-the-art power outage prediction models. The authors are currently exploring the possibility of beta regression as a power outage

prediction model. Accordingly, we have modified section 9 on "Suggested future research for "comprehensive" outage risk assessments" to point out that Beta distributions are just a suggestion (and we are currently working on that) as follows:

Efforts are still needed to overcome the limitations of state-of-the-art power outage prediction models. In this paper, we suggest the study of Beta GAMs to address them (Ferrari and Cribari-Neto, 2010; Olkin and Liu, 2003). While beta regression analysis was not the scope of this paper, we suggest possible candidate methods to overcome the challenges with state-of-the-art power outage prediction models. The authors are currently exploring the possibility of beta regression as a power outage prediction model. Beta distributions model random variables that take values from 0 to 1. Thus, it can model the fraction of outages in a city. For illustration, we present the possible prediction ranges of outages with wind speed for a Beta distributed fraction of outages (Yee, 2012; Olkin and Liu, 2003; Douma and Weedon, 2019). Similar to Negative Binomial GAM, Beta GAM can account for the high variability of the input variables and can handle the overdispersion in power outage data (Douma and Weedon, 2019).

Beta GAMs can make bounded predictions on the percentage of customers without electricity. Beta GAMs could extrapolate outages for the extreme (low and high) values of winds. Beta predictions always go from their minimum value of 0 to their maximum value of 1. Beta GAMs could predict the outages with low variance at low and high winds closer to the physics of infrastructure failure. Thus, future research could focus on developing such Beta GAMs for outage prediction.

Reference:

Ferrari, S. L. P., & Cribari-Neto, F. (2010). Beta Regression for Modelling Rates and Proportions. *Http://Dx.Doi.Org/10.1080/0266476042000214501*, *31*(7), 799–815. https://doi.org/10.1080/0266476042000214501

Olkin, I., & Liu, R. (2003). A bivariate beta distribution. *Statistics and Probability Letters*, *62*(4), 407–412. https://doi.org/10.1016/S0167-7152(03)00048-8

Yee, T. W. (2012). *Package "VGAM" (Vector generalized linear and additive models)*. http://www.springer.com/series/692

Douma, J. C., & Weedon, J. T. (2019). Analysing continuous proportions in ecology and evolution: A practical introduction to beta and Dirichlet regression. *Methods in Ecology and Evolution*, *10*(9), 1412–1430. https://doi.org/10.1111/2041-210X.13234

---

## Author Response (AR2)

**Response to Reviewer#1**

Thanks for working on revising the paper and addressing the issues that I have raised. And the quality of the paper has improved. However, I still found two additional issues that need to be addressed.

Dear Reviewer,

We thank you for reviewing our work, and your comments helped us to improve the manuscript. Please find the detailed response to these and other specific comments.

**Comment#1:** The discussion seems not very relevant to discoveries from this research but more like a wide discussion. So changes need to be made.

**Response#1:**

We agree with the reviewer that the discussion focused on general problems that could arise from erroneous power outage predictions. Thus, instead of keeping a separate discussion session, we have moved the contents of the discussion that are relevant just to the findings of this paper to the conclusion section. Please see the changes in Line 543 (also highlighted here in blue).

Line 543… Overestimated power outages could result in prioritizing a less affected city, placing more resources on that city than required. Limited mobility of crews during a disaster can lead to prolonged outages, delaying the restoration effects (National Academies of Sciences, Engineering, and Medicine, 2017). In general, erroneous power outage estimates with high uncertainty can result in the non-optimal placement of resources, as optimal resource allocation algorithms will use predicted outages (Brown, 2002).

Reference:

Brown, R. E. (2002). *Electric Power Distribution Reliability*. http://www.dekker.com
Enhancing the Resilience of the Nation's Electricity System. (2017). In *Enhancing the Resilience of the Nation's Electricity System*. https://doi.org/10.17226/24836

**Comment#2:** There is no background information about Beta BAMs and it seems to appear suddenly at the end of the paper. If you believe it works better, why it is not tested in the current study? I suggest reducing the amount of discussion of Beta BAMS.

**Response#2:**

Thank you for pointing this out. Since we did not test beta regression analysis as it was not the scope of this paper, we have removed the section on Beta GAMs. But, since we are currently exploring the possibility of different regression as a power outage prediction model, we suggested possible candidate methods to overcome the challenges with state-of-the-art power outage prediction models as the future research in the conclusion section. Accordingly, we have modified the last section to "Conclusions and Future Research" to point out that the additional models are just a suggestion (and we are currently working on that) as follows:

We suggest Beta and Binomial regressions to model power outages in future research. While testing their performance fell outside this paper's scope, Beta and Binomial distribution can help overcome existing limitations due to their fundamental properties. For example, Beta and Binomial regressions are upper-bounded, unlike Negative Binomial GLM and GAM regressions. Thus, Beta or Binomial GAMs can model the fraction of outages in a city, i.e., directly in the case of Beta since it goes from 0 to 1, or after normalizing the total number of outages by the maximum number of customers in the case of the Binomial regressions. Also, Beta and Binomial GAMs can extrapolate outages for the extreme (low and high) values of winds since they can model monotonically increasing outages as a function of environmental parameters, like winds. Finally, Beta and Binomial GAMs have variance closer to zero at outage fraction observations values of zero and 1, representing better the physics or power infrastructure failures.

Reference:

Dunn, P. K. and Smyth, G. K.: Generalized Linear Models With Examples in R, https://link.springer.com/book/10.1007/978-1-4419-0118-7, 2018.

Ferrari, S. L. and Cribari-Neto, F.: Beta Regression for Modelling Rates and Proportions, http://dx.doi.org/10.1080/0266476042000214501, 31, 799–815, https://doi.org/10.1080/0266476042000214501, 2010. 234